# A Contrastive Learning Approach for Training Variational Autoencoder Priors

**Jyoti Aneja**[1], **Alexander G. Schwing**[1], **Jan Kautz**[2], **Arash Vahdat**[2]
[1]University of Illinois at Urbana-Champaign, [2]NVIDIA
[1]{janeja2, aschwing}@illinois.edu, [2]{jkautz,avahdat}@nvidia.com

## Abstract

Variational autoencoders (VAEs) are one of the powerful likelihood-based generative models with applications in many domains. However, they struggle to generate high-quality images, especially when samples are obtained from the prior without any tempering. One explanation for VAEs' poor generative quality is the prior hole problem: the prior distribution fails to match the aggregate approximate posterior. Due to this mismatch, there exist areas in the latent space with high density under the prior that do not correspond to any encoded image. Samples from those areas are decoded to corrupted images. To tackle this issue, we propose an energy-based prior defined by the product of a base prior distribution and a reweighting factor, designed to bring the base closer to the aggregate posterior. We train the reweighting factor by noise contrastive estimation, and we generalize it to hierarchical VAEs with many latent variable groups. Our experiments confirm that the proposed noise contrastive priors improve the generative performance of state-of-the-art VAEs by a large margin on the MNIST, CIFAR-10, CelebA 64, and CelebA HQ 256 datasets. Our method is simple and can be applied to a wide variety of VAEs to improve the expressivity of their prior distribution.

## 1 Introduction

Variational autoencoders (VAEs) [39, 64] are one of the powerful likelihood-based generative models that have applications in image generation [6, 35, 62], music synthesis [12], speech generation [54, 60], image captioning [2, 3, 11], semi-supervised learning [33, 40], and representation learning [15, 79].

Although there has been tremendous progress in improving the expressivity of the approximate posterior, several studies have observed that VAE priors fail to match the *aggregate (approximate) posterior* [30, 66]. This phenomenon is sometimes described as *holes in the prior*, referring to regions in the latent space that are not decoded to data-like samples. Such regions often have a high density under the prior but have a low density under the aggregate approximate posterior.

The prior hole problem is commonly tackled by increasing the flexibility of the prior via hierarchical priors [42], autoregressive models [21], a mixture of encoders [72], normalizing flows [8, 81], resampled priors [5],

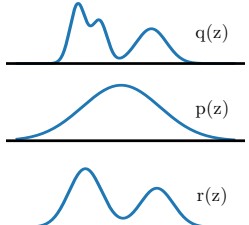

**Figure 1:** We propose an EBM prior using the product of a base prior $p(\mathbf{z})$ and a reweighting factor $r(\mathbf{z})$, designed to bring $p(\mathbf{z})$ closer to the aggregate posterior $q(\mathbf{z})$.

and energy-based models [57, 75–77]. Among them, energy-based models (EBMs) [13, 57] have shown promising results. However, they require running iterative MCMC during training which is computationally expensive when the energy function is represented by a neural network. Moreover, they scale poorly to hierarchical models where an EBM is defined on each group of latent variables.

35th Conference on Neural Information Processing Systems (NeurIPS 2021).

Our key insight in this work is that a trainable prior is brought as close as possible to the aggregate posterior as a result of training a VAE. The mismatch between the prior and the aggregate posterior can be reduced by reweighting the prior to re-adjust its likelihood in the area of mismatch with the aggregate posterior. To represent this reweighting mechanism, we formulate the prior using an EBM that is defined by the product of a reweighting factor and a base trainable prior as shown in Fig. 1. We represent the reweighting factor using neural networks and the base prior using Normal distributions.

Instead of computationally expensive MCMC sampling, notorious for being slow and often sensitive to the choice of parameters [13], we use noise contrastive estimation (NCE) [22] for training the EBM prior. We show that NCE trains the reweighting factor in our prior by learning a binary classifier to distinguish samples from a target distribution (i.e., approximate posterior) vs. samples from a noise distribution (i.e., the base trainable prior). However, since NCE's success depends on closeness of the noise distribution to the target distribution, we first train the VAE with the base prior to bring it close to the aggregate posterior. And then, we train the EBM prior using NCE.

In this paper, we make the following contributions: i) We propose an EBM prior termed *noise contrastive prior (NCP)* which is trained by contrasting samples from the aggregate posterior to samples from a base prior. NCPs are simple and can be learned as a post-training mechanism to improve the expressivity of the prior. ii) We also show how NCPs are trained on hierarchical VAEs with many latent variable groups. We show that training hierarchical NCPs scales easily to many groups, as they are trained for each latent variable group in parallel. iii) Finally, we demonstrate that NCPs improve the generative quality of several forms of VAEs by a large margin across datasets.

## 2 Related Work

In this section, we review related prior works.

**Energy-based Models (EBMs):** Early work on EBMs for generative learning goes back to the 1980s [1, 27]. Prior to the modern deep learning era, most attempts for building generative models using EBMs were centered around Boltzmann machines [26, 28] and their "deep" extensions [44, 67]. Although the energy function in these models is restricted to simple bilinear functions, they have been proven effective for representing the prior in discrete VAEs [65, 75–77]. Recently, EBMs with neural energy functions have gained popularity for representing complex data distributions [13]. Pang et al. [57] have shown that neural EBMs can represent expressive prior distributions. However, in this case, the prior is trained using MCMC sampling, and it has been limited to a single group of latent variables. VAEBM [80] combines the VAE decoder with an EBM defined on the pixel space and trains the model using MCMC. Additionally, VAEBM assumes that data lies in a continuous space and applies the energy function in that space. Hence, it cannot be applied to discrete data such as text or graphs. In contrast, NCP-VAE forms the energy function in the latent space and can be applied to non-continuous data. For continuous data, our model can be used along with VAEBM. We believe VAEBM and NCP-VAE are complementary. To avoid MCMC sampling, NCE [22] has recently been used for training a normalizing flow on data distributions [16]. Moreover, Han et al. [23, 24] use divergence triangulation to sidesteps MCMC sampling. In contrast, we use NCE to train an EBM prior where a noise distribution is easily available through a pre-trained VAE.

**Adversarial Training:** Similar to NCE, generative adversarial networks (GANs) [18] rely on a discriminator to learn the likelihood ratio between noise and real images. However, GANs use the discriminator to update the generator, whereas in NCE, the noise generator is fixed. In spirit similar are recent works [4, 7, 73] that link GANs, defined in the pixels space, to EBMs. We apply the likelihood ratio trick to the latent space of VAEs. The main difference: the base prior and approximate posterior are trained with the VAE objective rather than the adversarial loss. Adversarial loss has been used for training implicit encoders in VAEs [14, 51, 52]. But, they have not been linked to energy-based priors as we do explicitly.

**Prior Hole Problem:** Among prior works on this problem, VampPrior [72] uses a mixture of encoders to represent the prior. However, this requires storing training data or pseudo-data to generate samples at test time. Takahashi et al. [70] use the likelihood ratio estimator to train a simple prior distribution. However at test time, the aggregate posterior is used for sampling in the latent space.

**Reweighted Priors:** Bauer & Mnih [5] propose a reweighting factor similar to ours, but it is trained via truncated rejection sampling. Lawson et al. [45] introduce *energy-inspired models (EIMs)* that

define distributions induced by the sampling processes used by Bauer & Mnih [5] as well as our sampling-importance-resampling (SIR) sampling (called SNIS by Lawson et al. [45]). Although, EIMs have the advantage of end-to-end training, they require multiple samples during training (up to 1K). This can make application of EIMs to deep hierarchical models such as NVAEs very challenging as these models are memory intensive and are trained with a few training samples per GPU. Moreover, our NCP scales easily to hierarchical models where the reweighting factor for each group is trained in parallel with other groups (i.e., NCP enables model parallelism). We view our proposed training method as a simple alternative approach that allows us to scale up EBM priors to large VAEs.

**Two-stage VAEs:** VQ-VAE [62, 79] first trains an autoencoder and then fits an autoregressive PixelCNN [78] prior to the latent variables. Albeit impressive results, autoregressive models can be very slow to sample from. Two-stage VAE (2s-VAE) [10] trains a VAE on the data, and then, trains another VAE in the latent space. Regularized autoencoders (RAE) [17] train an autoencoder, and subsequently a Gaussian mixture model on latent codes. In contrast, we train the model with the original VAE objective in the first stage, and we improve the expressivity of the prior using an EBM.

## 3 Background

We first review VAEs, their extension to hierarchical VAEs before discussing the prior hole problem.

**Variational Autoencoders:** VAEs learn a generative distribution $p(\mathbf{x}, \mathbf{z}) = p(\mathbf{z})p(\mathbf{x}|\mathbf{z})$ where $p(\mathbf{z})$ is a prior distribution over the latent variable $\mathbf{z}$ and $p(\mathbf{x}|\mathbf{z})$ is a likelihood function that generates the data $\mathbf{x}$ given $\mathbf{z}$. VAEs are trained by maximizing a variational lower bound $\mathcal{L}_{\text{VAE}}(\mathbf{x})$ on the log-likelihood $\log p(\mathbf{x}) \geq \mathcal{L}_{\text{VAE}}(\mathbf{x})$ where

$$\mathcal{L}_{\text{VAE}}(\mathbf{x}) := \mathbb{E}_{q(\mathbf{z}|\mathbf{x})}[\log p(\mathbf{x}|\mathbf{z})] - \text{KL}(q(\mathbf{z}|\mathbf{x})||p(\mathbf{z})). \tag{1}$$

Here, $q(\mathbf{z}|\mathbf{x})$ is an approximate posterior and KL is the Kullback–Leibler divergence. The final training objective is $\mathbb{E}_{p_d(\mathbf{x})}[\mathcal{L}_{\text{VAE}}(\mathbf{x})]$ where $p_d(\mathbf{x})$ is the data distribution [39].

**Hierarchical VAEs (HVAEs):** To increase the expressivity of both prior and approximate posterior, earlier work adapted a hierarchical latent variable structure [3, 9, 19, 41, 68, 74]. In HVAEs, the latent variable $\mathbf{z}$ is divided into $K$ separate *groups*, $\mathbf{z} = \{\mathbf{z}_1, \dots, \mathbf{z}_K\}$. The approximate posterior and the prior distributions are then defined by $q(\mathbf{z}|\mathbf{x}) = \prod_{k=1}^{K} q(\mathbf{z}_k|\mathbf{z}_{<k}, \mathbf{x})$ and $p(\mathbf{z}) = \prod_{k=1}^{K} p(\mathbf{z}_k|\mathbf{z}_{<k})$. Using these, the training objective becomes

$$\mathcal{L}_{\text{HVAE}}(\mathbf{x}) := \mathbb{E}_{q(\mathbf{z}|\mathbf{x})}[\log p(\mathbf{x}|\mathbf{z})] - \sum_{k=1}^{K} \mathbb{E}_{q(\mathbf{z}_{<k}|\mathbf{x})}\left[\text{KL}(q(\mathbf{z}_k|\mathbf{z}_{<k}, \mathbf{x})||p(\mathbf{z}_k|\mathbf{z}_{<k}))\right], \tag{2}$$

where $q(\mathbf{z}_{<k}|\mathbf{x}) = \prod_{i=1}^{k-1} q(\mathbf{z}_i|\mathbf{z}_{<i}, \mathbf{x})$ is the approximate posterior up to the $(k-1)^{\text{th}}$ group.[1]

**Prior Hole Problem:** Let $q(\mathbf{z}) \triangleq \mathbb{E}_{p_d(\mathbf{x})}[q(\mathbf{z}|\mathbf{x})]$ denote the aggregate (approximate) posterior. In Appendix B.1, we show that maximizing $\mathbb{E}_{p_d(\mathbf{x})}[\mathcal{L}_{\text{VAE}}(\mathbf{x})]$ w.r.t. the prior parameters corresponds to bringing the prior as close as possible to the aggregate posterior by minimizing $\text{KL}(q(\mathbf{z})||p(\mathbf{z}))$ w.r.t. $p(\mathbf{z})$. Formally, the prior hole problem refers to the phenomenon that $p(\mathbf{z})$ fails to match $q(\mathbf{z})$.

## 4 Noise Contrastive Priors (NCPs)

One of the main causes of the prior hole problem is the limited expressivity of the prior that prevents it from matching the aggregate posterior. Recently, EBMs have shown promising results in representing complex distributions. Motivated by their success, we introduce the noise contrastive prior (NCP) $p_{\text{NCP}}(\mathbf{z}) = \frac{1}{Z} r(\mathbf{z})p(\mathbf{z})$, where $p(\mathbf{z})$ is a base prior distribution, e.g., a Normal, $r(\mathbf{z})$ is a reweighting factor, and $Z = \int r(\mathbf{z})p(\mathbf{z})d\mathbf{z}$ is the normalization constant. The function $r : \mathbb{R}^n \to \mathbb{R}^+$ maps the latent variable $\mathbf{z} \in \mathbb{R}^n$ to a positive scalar, and can be implemented using neural nets.

The reweighting factor $r(\mathbf{z})$ can be trained using MCMC as discussed in Appendix A. However, MCMC requires expensive sampling iterations that scale poorly to hierarchical VAEs. To address this, we describe a noise contrastive estimation based approach to train $p_{\text{NCP}}(\mathbf{z})$ without MCMC.

---

[1] For $k = 1$, the expectation inside the summation is simplified to $\text{KL}(q(\mathbf{z}_1|\mathbf{x})||p(\mathbf{z}_1))$.

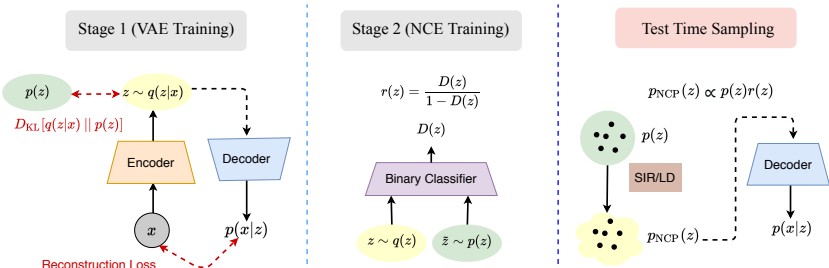

**Figure 2:** NCP-VAE is trained in two stages. In the first stage, we train a VAE using the original VAE objective. In the second stage, we train the reweighting factor $r(\mathbf{z})$ using noise contrastive estimation (NCE). NCE trains a classifier to distinguish samples from the prior and samples from the aggregate posterior. Our noise contrastive prior (NCP) is then constructed by the product of the base prior and the reweighting factor, formed via the classifier. At test time, we sample from NCP using sampling-importance-resampling (SIR) or Langevin dynamics (LD). These samples are then passed to the decoder to generate output samples.

### 4.1 Two-stage Training for Noise Contrastive Priors

To properly learn the reweighting factor, NCE training requires the base prior distribution to be close to the target distribution. To this end, we propose a two-stage training algorithm. In the first stage, we train the VAE with only the base prior $p(\mathbf{z})$. From Appendix B.1, we know that at the end of training, $p(\mathbf{z})$ is as close as possible to $q(\mathbf{z})$. In the second stage, we freeze the VAE model including the approximate posterior $q(\mathbf{z}|\mathbf{x})$, the base prior $p(\mathbf{z})$, and the likelihood $p(\mathbf{x}|\mathbf{z})$, and we only train the reweighting factor $r(\mathbf{z})$. This second stage can be thought of as replacing the base distribution $p(\mathbf{z})$ with a more expressive distribution of the form $p_{\text{NCP}}(\mathbf{z}) \propto r(\mathbf{z})p(\mathbf{z})$. Note that our proposed method is generic as it only assumes that we can draw samples from $q(\mathbf{z})$ and $p(\mathbf{z})$, which applies to any VAE. This proposed training is illustrated in Fig. 2. Next, we present our approach for training $r(\mathbf{z})$.

### 4.2 Learning The Reweighting Factor with Noise Contrastive Estimation

Recall that maximizing the variational bound in Eq. 1 with respect to the prior's parameters corresponds to closing the gap between the prior and the aggregate posterior by minimizing $\text{KL}(q(\mathbf{z})||p_{\text{NCP}}(\mathbf{z}))$ with respect to the prior $p_{\text{NCP}}(\mathbf{z})$. Assuming that the base $p(\mathbf{z})$ in $p_{\text{NCP}}(\mathbf{z})$ is fixed after the first stage, $\text{KL}(q(\mathbf{z})||p_{\text{NCP}}(\mathbf{z}))$ is zero when $r(\mathbf{z}) = q(\mathbf{z})/p(\mathbf{z})$. However, since we do not have the density function for $q(\mathbf{z})$, we cannot compute the ratio explicitly. Instead, in this paper, we propose to estimate $r(\mathbf{z})$ using noise contrastive estimation [22], also known as the likelihood ratio trick, that has been popularized in machine learning by predictive coding [55] and generative adversarial networks (GANs) [18]. Since, we can generate samples from both $p(\mathbf{z})$ and $q(\mathbf{z})^2$, we train a binary classifier to distinguish samples from $q(\mathbf{z})$ and samples from the base prior $p(\mathbf{z})$ by minimizing the binary cross-entropy loss

$$\min_D - \mathbb{E}_{\mathbf{z}\sim q(\mathbf{z})}[\log D(\mathbf{z})] - \mathbb{E}_{\mathbf{z}\sim p(\mathbf{z})}[\log(1 - D(\mathbf{z}))]. \tag{3}$$

Here, $D : \mathbb{R}^n \to (0, 1)$ is a binary classifier that generates the classification prediction probabilities. Eq. (3) is minimized when $D(\mathbf{z}) = \frac{q(\mathbf{z})}{q(\mathbf{z})+p(\mathbf{z})}$. Denoting the classifier at optimality by $D^*(\mathbf{z})$, we estimate the reweighting factor $r(\mathbf{z}) = \frac{q(\mathbf{z})}{p(\mathbf{z})} \approx \frac{D^*(\mathbf{z})}{1-D^*(\mathbf{z})}$. The appealing advantage of this estimator is that it is obtained by simply training a binary classifier rather than using expensive MCMC sampling.

Intuitively, if $p(\mathbf{z})$ is very close to $q(\mathbf{z})$ (i.e., $p(\mathbf{z}) \approx q(\mathbf{z})$), the optimal classifier will have a large loss value in Eq. (3), and we will have $r(\mathbf{z}) \approx 1$. If $p(\mathbf{z})$ is instead far from $q(\mathbf{z})$, the binary classifier will easily learn to distinguish samples from the two distributions and it will not learn the likelihood ratios correctly. If $p(\mathbf{z})$ is roughly close to $q(\mathbf{z})$, then the binary classifier can learn the ratios successfully.

---

[2] We generate samples from the aggregate posterior $q(\mathbf{z}) = \mathbb{E}_{p_d(\mathbf{x})}[q(\mathbf{z}|\mathbf{x})]$ via ancestral sampling: draw data from the training set ($\mathbf{x} \sim p_d(\mathbf{x})$) and then sample from $\mathbf{z} \sim q(\mathbf{z}|\mathbf{x})$.

### 4.3 Test Time Sampling

To sample from a VAE with an NCP, we first generate samples from the NCP and pass them to the decoder to generate output samples (Fig. 2). We propose two methods for sampling from NCPs.

**Sampling-Importance-Resampling (SIR):** We first generate $M$ samples from the base prior distribution $\{\mathbf{z}^{(m)}\}_{m=1}^M \sim p(\mathbf{z})$. We then resample one of the $M$ proposed samples using importance weights proportional to $w^{(m)} = p_{\text{NCP}}(\mathbf{z}^{(m)})/p(\mathbf{z}^{(m)}) = r(\mathbf{z}^{(m)})$. The benefit of this technique: both proposal generation and the evaluation of $r$ on the samples are done in parallel.

**Langevin Dynamics (LD):** Since our NCP is an EBM, we can use LD for sampling. Denoting the energy function by $E(\mathbf{z}) = -\log r(\mathbf{z}) - \log p(\mathbf{z})$, we initialize a sample $\mathbf{z}_0$ by drawing from $p(\mathbf{z})$ and update the sample iteratively using: $\mathbf{z}_{t+1} = \mathbf{z}_t - 0.5\,\lambda\nabla_{\mathbf{z}}E(\mathbf{z}) + \sqrt{\lambda}\epsilon_t$ where $\epsilon_t \sim \mathcal{N}(0,1)$ and $\lambda$ is the step size. LD is run for a finite number of iterations, and in contrast to SIR, it is slower given its sequential form.

### 4.4 Generalization to Hierarchical VAEs

The state-of-the-art VAEs [9, 74] use a hierarchical $q(\mathbf{z}|\mathbf{x})$ and $p(\mathbf{z})$. Here $p(\mathbf{z})$ is chosen to be a Gaussian distribution. Appendix B.2 shows that training a HVAE encourages the prior to minimize $\mathbb{E}_{q(\mathbf{z}_{<k})}[\text{KL}(q(\mathbf{z}_k|\mathbf{z}_{<k})||p(\mathbf{z}_k|\mathbf{z}_{<k}))]$ for each conditional, where $q(\mathbf{z}_{<k}) \triangleq \mathbb{E}_{p_d(\mathbf{x})}[q(\mathbf{z}_{<K}|\mathbf{x})]$ is the aggregate posterior up to the $(k-1)^{\text{th}}$ group, and $q(\mathbf{z}_k|\mathbf{z}_{<k}) \triangleq \mathbb{E}_{p_d(\mathbf{x})}[q(\mathbf{z}_k|\mathbf{z}_{<k},\mathbf{x})]$ is the aggregate conditional for the $k^{\text{th}}$ group. Given this observation, we extend NCPs to hierarchical models to match each conditional in the prior with $q(\mathbf{z}_k|\mathbf{z}_{<k})$. Formally, we define hierarchical NCPs by $p_{\text{NCP}}(\mathbf{z}) = \frac{1}{Z}\prod_{k=1}^K r(\mathbf{z}_k|\mathbf{z}_{<k})p(\mathbf{z}_k|\mathbf{z}_{<k})$ where each factor is an EBM. $p_{\text{NCP}}(\mathbf{z})$ resembles EBMs with autoregressive structure among groups [53].

In the first stage, we train the HVAE with prior $\prod_{k=1}^K p(\mathbf{z}_k|\mathbf{z}_{<k})$. For the second stage, we use $K$ binary classifiers, each for a hierarchical group. Following Appendix C, we train each classifier via

$$\min_{D_k} \mathbb{E}_{p_d(\mathbf{x})q(\mathbf{z}_{<k}|\mathbf{x})}\Big[- \mathbb{E}_{q(\mathbf{z}_k|\mathbf{z}_{<k},\mathbf{x})}[\log D_k(\mathbf{z}_k,c(\mathbf{z}_{<k}))] - \mathbb{E}_{p(\mathbf{z}_k|\mathbf{z}_{<k})}[\log(1 - D_k(\mathbf{z}_k,c(\mathbf{z}_{<k})))]\Big], \tag{4}$$

where the outer expectation samples from groups up to the $(k-1)^{\text{th}}$ group, and the inner expectations sample from approximate posterior and base prior for the $k^{\text{th}}$ group, conditioned on the same $\mathbf{z}_{<k}$. The discriminator $D_k$ classifies samples $\mathbf{z}_k$ while conditioning its prediction on $\mathbf{z}_{<k}$ using a shared context feature $c(\mathbf{z}_{<k})$.

The NCE training in Eq. (4) is minimized when $D_k(\mathbf{z}_k,c(\mathbf{z}_{<k})) = \frac{q(\mathbf{z}_k|\mathbf{z}_{<k})}{q(\mathbf{z}_k|\mathbf{z}_{<k})+p(\mathbf{z}_k|\mathbf{z}_{<k})}$. Denoting the classifier at optimality by $D_k^*(\mathbf{z},c(\mathbf{z}_{<k}))$, we obtain the reweighting factor $r(\mathbf{z}_k|\mathbf{z}_{<k}) \approx \frac{D_k^*(\mathbf{z}_k,c(\mathbf{z}_{<k}))}{1-D_k^*(\mathbf{z}_k,c(\mathbf{z}_{<k}))}$ in the second stage. Given our hierarchical NCP, we use ancestral sampling to sample from the prior. For sampling from each group, we can use SIR or LD as discussed before.

The context feature $c(\mathbf{z}_{<k})$ extracts a representation from $\mathbf{z}_{<k}$. Instead of learning a new representation at stage two, we simply use the representation that is extracted from $\mathbf{z}_{<k}$ in the hierarchical prior, trained in the first stage. Note that the binary classifiers are trained in parallel for all groups.

## 5 Experiments

In this section, we situate NCP against prior work on several commonly used single group VAE models in Sec. 5.1. In Sec. 5.2, we present our main results where we apply NCP to hierarchical NVAE [74] to demonstrate that our approach can be applied to large scale models successfully.

In most of our experiments, we measure the sample quality using the Fréchet Inception Distance (FID) score [25] with 50,000 samples, as computing the log-likelihood requires estimating the intractable normalization constant. For generating samples from the model, we use SIR with 5K proposal samples. To report log-likelihood results, we train models with small latent spaces on the dynamically binarized MNIST [46] dataset. We intentionally limit the latent space to ensure that we can estimate the normalization constant correctly.

**Table 1:** Comparison with two-stage VAEs on CelebA-64 with RAE [17] networks. [†] Results reported by Ghosh et al. [17].

| Model | FID↓ |
|---|---|
| VAE w/ Gaussian prior | 48.12[†] |
| 2s-VAE [10] | 49.70[†] |
| WAE [71] | 42.73[†] |
| RAE [17] | 40.95[†] |
| NCP w/ Gaussian prior as base | 41.28 |
| NCP w/ GMM prior as base | **39.00** |
| Base VAE-Recon | 36.01 |

**Table 2:** Likelihood results on MNIST on single latent group model with architecture from LARS [5] & SNIS [45] (results in nats). We closely follow the training hyperparameters used by Lawson et al. [45].

| Model | NLL↓ |
|---|---|
| VAE w/ Gaussian prior | 84.82 |
| VAE w/ LARS prior [5] | 83.03 |
| VAE w/ SNIS prior [45] | **82.52** |
| NCP-VAE | 82.82 |

## 5.1 Comparison to Prior Work

In this section, we apply NCP to several commonly used small VAE models. Our goal, here, is to situate our proposed model against (i) two-stage VAE models that train a (variational) autoencoder first, and then, fit a prior distribution (Sec. 5.1.1), and (ii) VAEs with reweighted priors (Sec. 5.1.2). To make sure that these comparisons are fair, we follow exact training setup and network architectures from prior work as discussed below.

### 5.1.1 Comparison against Two-Stage VAEs

Here, we show the generative performance of our approach applied to the VAE architecture in RAE [17] on the CelebA-64 dataset [48]. We borrow the exact training setup from [17] and implement our method using their publicly available code.[3] Note that this VAE architecture has only one latent variable group. The same base architecture was used in the implementation of 2s-VAE [10] and WAE [71]. In order to compare our method to these models, we use the reported results from RAE [17]. We apply our NCP-VAE on top of both vanilla VAE with a Gaussian prior and a 10-component Gaussian mixture model (GMM) prior that was proposed in RAEs. As we can see in Tab. 1, our NCP-VAE improves the performance of the base VAE, improving the FID score to 41.28 from 48.12. Additionally, when NCP is applied to the VAE with GMM prior (the RAE model), it improves its performance from 40.95 to the FID score of 39.00. We also report the FID score for reconstructed images using samples from the aggregate posterior $q(\mathbf{z})$ instead of the prior. Note that this value represents the best FID score that can be obtained by perfectly matching the prior to the aggregate posterior in the second stage. The high FID score of 36.01 indicates that the small VAEs cannot reconstruct data samples well due to the small network architecture and latent space. Thus, even with expressive priors, FID for two-stage VAEs are lower bounded by 36.01 in the 2nd stage.

### 5.1.2 Comparison against Reweighted Priors

LARS [5] and SNIS [45] train reweighted priors similar to our EBM prior. To compare NCP-VAE against these methods, we implement our method using the VAE and energy-function networks from [45]. We closely follow the training hyperparameters used in [45] as well as their approach for obtaining a lower bound on the log likelihood (i.e., the SNIS objective in [45] provides a lower bound on data likelihood). As shown in Tab. 2, NCP-VAE obtains the negative log-likelihood (NLL) of 82.82, comparable to Lawson et al. [45], while outperforming LARS [5]. Although NCP-VAE is slightly inferior to SNIS on MNIST, it has several advantages as discussed in Sec. 2.

### 5.1.3 Training using Normalizing Flows

Chen et al. [8] (Sec. 3.2) show that a normalizing flow in the approximate posterior is equivalent to having its inverse in the prior. The base NVAE uses normalizing flows in the encoder. As a part of VAE training, prior and aggregate posterior are brought close, i.e., normalizing flows are implicitly used. We argue that normalizing flows provide limited gains to address the prior-hole problem (see Fig. 1 by Kingma et al. [41]). Yet, our model further improves the base VAE equipped with normalizing flow.

---

[3] `https://github.com/ParthaEth/Regularized_autoencoders-RAE-`

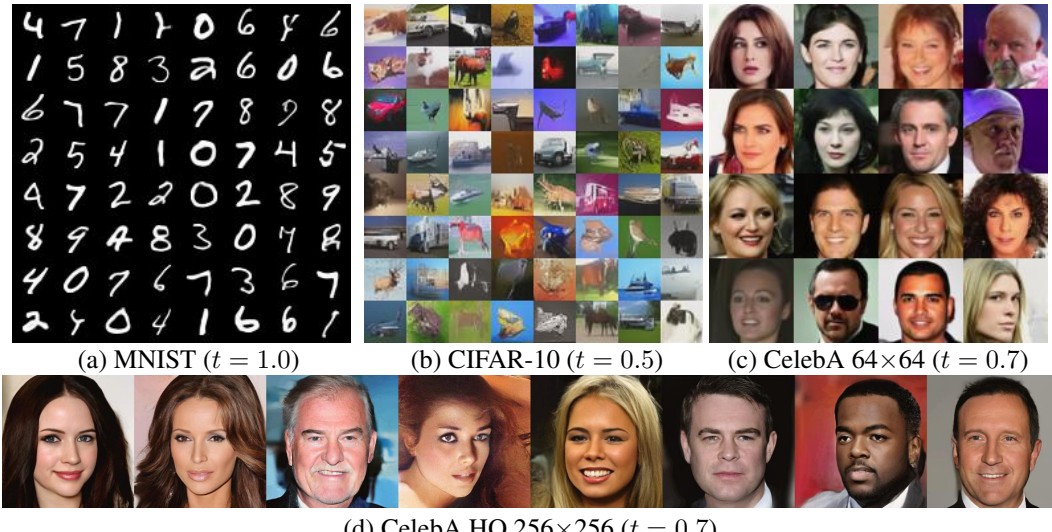

| (a) MNIST ($t = 1.0$) | (b) CIFAR-10 ($t = 0.5$) | (c) CelebA 64×64 ($t = 0.7$) |

(d) CelebA HQ 256×256 ($t = 0.7$)

**Figure 3:** Randomly sampled images from NCP-VAE with the temperature $t$ for the prior.

**Table 3:** Generative performance on CelebA-64.

| Model | FID↓ |
|---|---|
| NCP-VAE (ours) | **5.25** |
| VAEBM [80] | 5.31 |
| NVAE [74] | 13.48 |
| RAE [17] | 40.95 |
| 2s-VAE [10] | 44.4 |
| WAE [71] | 35 |
| Perceptial AE[82] | 13.8 |
| Latent EBM [57] | 37.87 |
| COCO-GAN [47] | 4.0 |
| QA-GAN [58] | 6.42 |
| NVAE-Recon [74] | 1.03 |

**Table 4:** Generative performance on CIFAR-10.

| Model | FID↓ |
|---|---|
| NCP-VAE (ours) | 24.08 |
| VAEBM [80] | **12.96** |
| NVAE [74] | 51.71 |
| RAE [17] | 74.16 |
| 2s-VAE [10] | 72.9 |
| Perceptial AE [82] | 51.51 |
| EBM [13] | 40.58 |
| Latent EBM [57] | 70.15 |
| Style-GANv2 [36] | 3.26 |
| DDPM [29] | 3.17 |
| Score SDE [69] | 3.20 |
| NVAE-Recon [74] | 2.67 |

## 5.2 Quantitative Results on Hierarchical Models

In this section, we apply NCP to the hierarchical VAE model proposed in NVAE [74]. We examine NCP-VAE on four datasets including dynamically binarized MNIST [46], CIFAR-10 [43], CelebA-64 [48] and CelebA-HQ-256 [34]. For CIFAR-10 and CelebA-64, the model has 30 groups, and for CelebA-HQ-256 it has 20 groups. For MNIST, we train an NVAE model with a small latent space on MNIST with 10 groups of $4 \times 4$ latent variables. The small latent space allows us to estimate the partition function confidently (std. of $\log Z$ estimation $\leq 0.23$). The quantitative results are reported in Tab. 3, Tab. 4, Tab. 5, and Tab. 6. On all four datasets, our model improves upon NVAE, and it reduces the gap with GANs by a large margin. On CelebA-64, we improve NVAE from an FID of 13.48 to 5.25, comparable to GANs. On CIFAR-10, NCP-VAE improves the NVAE FID of 51.71 to 24.08. On MNIST, although our latent space is much smaller, our model outperforms previous VAEs. NVAE has reported 78.01 nats on this dataset with a larger latent space.

On CIFAR-10 and CelebA-HQ-256, recently proposed VAEBM [80] outperforms our NCP-VAE. However, we should note that (i) NCP-VAE and VAEBM are complementary to each other, as NCP-VAE targets the latent space while VAEBM forms an EBM on the data space. We expect improvements by combining these two models. (ii) VAEBM assumes that the data lies on a continuous space whereas NCP-VAE does not make any such assumption and it can be applied to discrete data (like binarized MNIST in Tab. 6), graphs, and text. (iii) NCP-VAE is much simpler to setup as it involves training a binary classifier whereas VAEBM requires MCMC for both training and test.

**Table 5:** Generative results on CelebA-HQ-256.

| Model | FID↓ |
|---|---|
| NCP-VAE (ours) | 24.79 |
| VAEBM [80] | **20.38** |
| NVAE [74] | 40.26 |
| GLOW [38] | 68.93 |
| Advers. LAE [59] | 19.21 |
| PGGAN [34] | 8.03 |
| NVAE-Recon [74] | 0.45 |

**Table 6:** Likelihood results on MNIST in nats.

| Model | NLL↓ |
|---|---|
| NCP-VAE (ours) | **78.10** |
| NVAE-small [74] | 78.67 |
| BIVA [50] | 78.41 |
| DAVE++ [76] | 78.49 |
| IAF-VAE [41] | 79.10 |
| VampPrior AR dec. ([72]) | 78.45 |
| DVAE [65] | 80.15 |

## 5.3 Qualitative Results

We visualize samples generated by NCP-VAE with the NVAE backbone in Fig. 3 without any manual intervention. We adopt the common practice of reducing the temperature of the base prior $p(\mathbf{z})$ by scaling down the standard-deviation of the conditional Normal distributions [38].[4] [6, 74] also observe that re-adjusting the batch-normalization (BN), given a temperature applied to the prior, improves the generative quality. Similarly, we achieve diverse, high-quality images by re-adjusting the BN statistics as described by [74]. Additional qualitative results are shown in Appendix G.

**Nearest Neighbors from the Training Dataset:** To highlight that hierarchical NCP generates unseen samples at test time rather than memorizing the training dataset, Figures 4-7 visualize samples from the model along with a few training images that are most similar to them (nearest neighbors). To get the similarity score for a pair of images, we downsample to $64 \times 64$, center crop to $40 \times 40$ and compute the Euclidean distance. The KD-tree algorithm is used to fetch the nearest neighbors. We note that the generated samples are quite distinct from the training images.

Query Image     Nearest neighbors from the training dataset    .

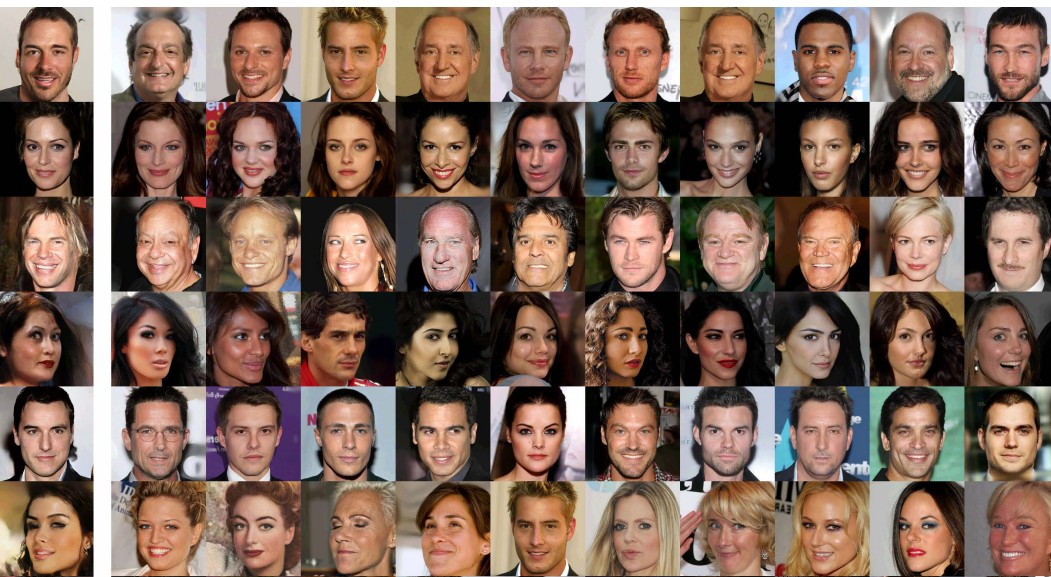

**Figure 4:** Query images (left) and their nearest neighbors from the CelebA-HQ-256 training dataset.

## 5.4 Additional Ablation Studies

We perform additional experiments to study i) how hierarchical NCPs perform as the number of latent groups increases, ii) the impact of SIR and LD hyperparameters, and iii) what the classification loss in NCE training conveys about $p(\mathbf{z})$ and $q(\mathbf{z})$. All experiments are performed on CelebA-64.

**Number of latent variable groups:** Tab. 7 shows the generative performance of hierarchical NCP with different amounts of latent variable groups. As we increase the number of groups, the FID

---

[4]Lowering the temperature is only used to obtain qualitative samples, not for the quantitative results in Sec. 5.

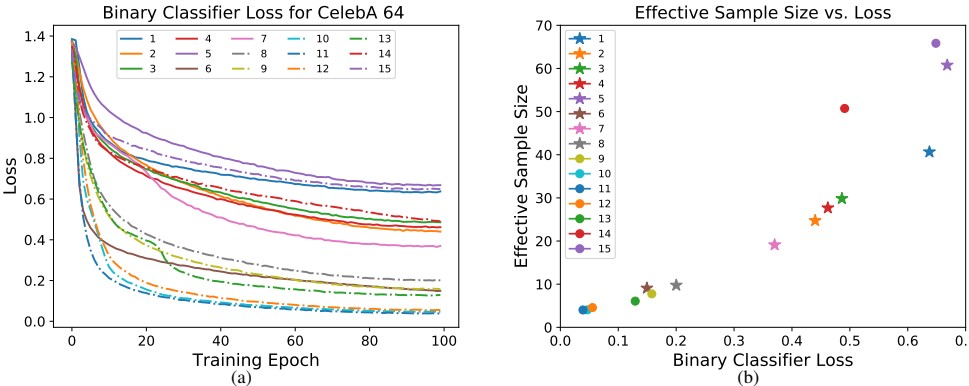

**Figure 5:** **(a)** Classification loss for binary classifiers on latent variable groups. A larger final loss upon training indicates that $q(\mathbf{z})$ and $p(\mathbf{z})$ are more similar. **(b)** The effective sample size vs. the final loss value at the end of training. Higher effective sample size implies similarity of two distributions.

score of both NVAE and our model improves. This shows the efficacy of our NCPs, with expressive hierarchical priors in the presence of many groups.

**SIR and LD parameters:** The computational complexity of SIR is similar to LD if we set the number of proposal samples in SIR equal to the number LD iterations. In Tab. 8, we observe that increasing both the number of proposal samples in SIR and the LD iterations leads to a noticeable improvement in FID score. For SIR, the proposal generation and the evaluation of $r(\mathbf{z})$ are parallelizable. Hence, as shown in Tab. 8, image generation is faster with SIR than with LD. However, GPU

**Table 7:** # groups & generative performance in FID↓.

| # groups | NVAE | NCP-VAE |
|---|---|---|
| 6 | 33.18 | 18.68 |
| 15 | 14.96 | 5.96 |
| 30 | 13.48 | **5.25** |

memory usage scales with the number of SIR proposals, but not with the number of LD iterations. Interestingly, SIR, albeit simple, performs better than LD when using about the same compute.

**Classification loss in NCE:** We can draw a direct connection between the classification loss in Eq. (3) and the similarity of $p(\mathbf{z})$ and $q(\mathbf{z})$. Denoting the classification loss in Eq. (3) at optimality by $\mathcal{L}^*$, Goodfellow et al. [18] show that $\text{JSD}(p(\mathbf{z})\|q(\mathbf{z})) = \log 2 - 0.5 \times \mathcal{L}^*$ where JSD denotes the Jensen–Shannon divergence between two distributions. Fig. 5(a) plots the classification loss (Eq. (4)) for each classifier for a 15-group NCP trained on the CelebA-64 dataset. Assume that the classifier loss at the end of training is a good approximation of $\mathcal{L}^*$. We observe that 8 out of 15 groups have $\mathcal{L}^* \geq 0.4$, indicating a good overlap between $p(\mathbf{z})$ and $q(\mathbf{z})$ for those groups. To further assess the impact of the distribution match on SIR sampling, in Fig. 5(b), we visualize the effective sample size (ESS)[5] in SIR vs. $\mathcal{L}^*$ for the same group. We observe a strong correlation between $\mathcal{L}^*$ and the effective sample size. SIR is more reliable on the same 8 groups that have high classification loss. These groups are at the top of the NVAE hierarchy which have been shown to control the global structure of generated samples (see B.6 in [74]).

**Table 8:** Effect of SIR sample size and LD iterations. Time-$N$ is the time used to generate a batch of $N$ images.

| # SIR proposal samples | FID↓ | Time-1 (sec) | Time-10 (sec) | Memory (GB) | # LD iterations | FID↓ | Time-1 (sec) | Time-10 (sec) | Memory (GB) |
|---|---|---|---|---|---|---|---|---|---|
| 5 | 11.75 | 0.34 | 0.42 | 1.96 | 5 | 14.44 | 3.08 | 3.07 | 1.94 |
| 50 | 8.58 | 0.40 | 1.21 | 4.30 | 50 | 12.76 | 27.85 | 28.55 | 1.94 |
| 500 | 6.76 | 1.25 | 9.43 | 20.53 | 500 | 8.12 | 276.13 | 260.35 | 1.94 |
| 5000 | **5.25** | 10.11 | 95.67 | 23.43 | 1000 | **6.98** | 552 | 561.44 | 1.94 |

**Analysis of the re-weighting technique:** To show that samples from NCP ($p_{\text{NCP}}(\mathbf{z})$) are closer to the aggregate posterior $q(\mathbf{z})$ compared to the samples from the base prior $p(\mathbf{z})$, we take 5k samples from $q(\mathbf{z})$, $p(\mathbf{z})$, and $p_{\text{NCP}}(\mathbf{z})$ at different hierarchy/group levels. Samples are projected to a lower dimension ($d = 500$) using PCA and populations are compared via Maximum Mean Discrepancy (MMD). Consistent with Fig. 5(a),

**Table 9:** MMD comparison.

| # group | $(q, p)$ | $(q, p_{\text{NCP}})$ |
|---|---|---|
| 5 | 0.002 | 0.002 |
| 10 | 0.08 | 0.06 |
| 12 | 0.08 | 0.07 |

---

[5]ESS measures reliability of SIR via $1/\sum_m (\hat{w}^{(m)})^2$, where $\hat{w}^{(m)} = r(\mathbf{z}^{(m)})/\sum_{m'} r(\mathbf{z}^{(m')})$ [56].

Tab. 9 shows that groups with lower classification loss had a mismatch between $p$ and $q$, and NCP is able to reduce the dissimilarity by re-weighting.

## 6 Conclusions

The prior hole problem is one of the main reasons for VAEs' poor generative quality. In this paper, we tackled this problem by introducing the noise contrastive prior (NCP), defined by the product of a reweighting factor and a base prior. We showed how the reweighting factor is trained by contrasting samples from the aggregate posterior with samples from the base prior. Our proposal is simple and can be applied to any VAE to increase its prior's expressivity. We also showed how NCP training scales to large hierarchical VAEs, as it can be done in parallel simultaneously for all the groups. Finally, we demonstrated that NCPs improve the generative performance of small single group VAEs and state-of-the-art NVAEs by a large margin.

## 7 Impact Statement

The main contributions of this paper are towards tackling a fundamental issue with training VAE models – *the prior hole problem*. The proposed method increases the expressivity of the distribution used to sample latent codes for test-time image generation, thereby increasing the quality (sharpness) and diversity of the generated solutions. Therefore, ideas from this paper could find applications in VAE-based content generation domains, such as computer graphics, biomedical imaging, computation fluid dynamics, among others. More generally, we expect the improved data generation to be beneficial for data augmentation and representation learning techniques.

When generating new content from a trained VAE model, one must carefully assess if the sampled distribution bears semblance to the real data distribution used for training, in terms of capturing the different modes of the real data, as well as the long tail. A model that fails to achieve a real data distribution result should be considered biased and corrective steps should be taken to proactively address it. Methods such as NCP-VAE, that increase the prior expressivity, hold the promise to reduce the bias in image VAEs. Even so, factors such as the VAE architecture, training hyper-parameters, and temperature for test-time generation, could impact the potential for bias and ought to be given due consideration. We recommend incorporating the active research into bias correction for generative modeling [20] into any potential applications that use this work.

## 8 Acknowledgements

The authors would like to thank Zhisheng Xiao for helpful discussions. They also would like to extend their sincere gratitude to the NGC team at NVIDIA for their compute support.

## 9 Funding Transparency Statement

This work was mostly funded by NVIDIA during an internship. It was also partially supported by NSF under Grant #1718221, 2008387, 2045586, 2106825, MRI #1725729, and NIFA award 2020-67021-32799.

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
