## A   Training Energy-based Priors using MCMC

In this section, we show how a VAE with energy-based model in its prior can be trained. Assuming that the prior is in the form $p_{\text{EBM}}(\mathbf{z}) = \frac{1}{Z} r(\mathbf{z}) p(\mathbf{z})$, the variational bound is of the form:

$$
\begin{aligned}
\mathbb{E}_{p_d(\mathbf{x})}[\mathcal{L}_{\text{VAE}}] &= \mathbb{E}_{p_d(\mathbf{x})}\left[\mathbb{E}_{q(\mathbf{z}|\mathbf{x})}[\log p(\mathbf{x}|\mathbf{z})] - \text{KL}(q(\mathbf{z}|\mathbf{x})||p_{\text{EBM}}(\mathbf{z}))\right] \\
&= E_{p_d(\mathbf{x})}\left[\mathbb{E}_{q(\mathbf{z}|\mathbf{x})}[\log p(\mathbf{x}|\mathbf{z}) - \log q(\mathbf{z}|\mathbf{x}) + \log r(\mathbf{z}) + \log p(\mathbf{z})]\right] - \log Z,
\end{aligned}
$$

where the expectation term, similar to VAEs, can be trained using the reparameterization trick. The only problematic term is the log-normalization constant $\log Z$, which captures the gradient with respect to the parameters of the prior $p_{\text{EBM}}(\mathbf{z})$. Denoting these parameters by $\theta$, the gradient of $\log Z$ is obtained by:

$$
\frac{\partial}{\partial \theta} \log Z = \frac{1}{Z} \int \frac{\partial (r(\mathbf{z}) p(\mathbf{z}))}{\partial \theta} d\mathbf{z} = \int \frac{r(\mathbf{z}) p(\mathbf{z})}{Z} \frac{\partial \log(r(\mathbf{z}) p(\mathbf{z}))}{\partial \theta} d\mathbf{z} = \mathbb{E}_{P_{EBM}(\mathbf{z})}\left[\frac{\partial \log(r(\mathbf{z}) p(\mathbf{z}))}{\partial \theta}\right],
\tag{5}
$$

where the expectation can be estimated using MCMC sampling from the EBM prior.

## B   Maximizing the Variational Bound from the Prior's Perspective

In this section, we discuss how maximizing the variational bound in VAEs from the prior's perspective corresponds to minimizing a KL divergence from the aggregate posterior to the prior. Note that this relation has been explored by Hoffman & Johnson [30], Rezende & Viola [63], Tomczak & Welling [72] and we include it here for completeness.

### B.1   VAE with a Single Group of Latent Variables

Denote the aggregate (approximate) posterior by $q(\mathbf{z}) \triangleq \mathbb{E}_{p_d(\mathbf{x})}[q(\mathbf{z}|\mathbf{x})]$. Here, we show that maximizing the $\mathbb{E}_{p_d(\mathbf{x})}[\mathcal{L}_{\text{VAE}}(\mathbf{x})]$ with respect to the prior parameters corresponds to learning the prior by minimizing $\text{KL}(q(\mathbf{z})||p(\mathbf{z}))$. To see this, note that the prior $p(\mathbf{z})$ only participates in the KL term in $\mathcal{L}_{\text{VAE}}$ (Eq. 1). We hence have:

$$
\begin{aligned}
\arg\max_{p(\mathbf{z})} \mathbb{E}_{p_d(\mathbf{x})}[\mathcal{L}_{\text{VAE}}(\mathbf{x})] &= \arg\min_{p(\mathbf{z})} \mathbb{E}_{p_d(\mathbf{x})}[\text{KL}(q(\mathbf{z}|\mathbf{x})||p(\mathbf{z}))] \\
&= \arg\min_{p(\mathbf{z})} -\mathbb{E}_{p_d(\mathbf{x})}[H(q(\mathbf{z}|\mathbf{x}))] - \mathbb{E}_{q(\mathbf{z})}[\log p(\mathbf{z})] \\
&= \arg\min_{p(\mathbf{z})} -H(q(\mathbf{z})) - \mathbb{E}_{q(\mathbf{z})}[\log p(\mathbf{z})] \\
&= \arg\min_{p(\mathbf{z})} \text{KL}(q(\mathbf{z})||p(\mathbf{z})),
\end{aligned}
$$

where $H(.)$ denotes the entropy. Above, we replaced the expected entropy $\mathbb{E}_{p_d(\mathbf{x})}[H(q(\mathbf{z}|\mathbf{x}))]$ with $H(q(\mathbf{z}))$ as the minimization is with respect to the parameters of the prior $p(\mathbf{z})$.

### B.2   Hierarchical VAEs

Denote hierarchical approximate posterior and prior distributions by: $q(\mathbf{z}|\mathbf{x}) = \prod_{k=1}^{K} q(\mathbf{z}_k|\mathbf{z}_{<k}, \mathbf{x})$ and $p(\mathbf{z}) = \prod_{k=1}^{K} p(\mathbf{z}_k|\mathbf{z}_{<k})$. The hierarchical VAE objective becomes:

$$
\mathcal{L}_{\text{HVAE}}(\mathbf{x}) = \mathbb{E}_{q(\mathbf{z}|\mathbf{x})}[\log p(\mathbf{x}|\mathbf{z})] - \sum_{k=1}^{K} \mathbb{E}_{q(\mathbf{z}_{<k}|\mathbf{x})}\left[\text{KL}(q(\mathbf{z}_k|\mathbf{z}_{<k}, \mathbf{x})||p(\mathbf{z}_k|\mathbf{z}_{<k}))\right],
\tag{6}
$$

where $q(\mathbf{z}_{<k}|\mathbf{x}) = \prod_{i=1}^{k-1} q(\mathbf{z}_i|\mathbf{z}_{<i}, \mathbf{x})$ is the approximate posterior up to the $(k-1)^{\text{th}}$ group. Denote the aggregate posterior up to the $(K-1)^{\text{th}}$ group by $q(\mathbf{z}_{<k}) \triangleq \mathbb{E}_{p_d(\mathbf{x})}[q(\mathbf{z}_{<K}|\mathbf{x})]$ and the aggregate conditional for the $k^{\text{th}}$ group given the previous groups $q(\mathbf{z}_k|\mathbf{z}_{<k}) \triangleq \mathbb{E}_{p_d(\mathbf{x})}[q(\mathbf{z}_k|\mathbf{z}_{<k}, \mathbf{x})]$.

Here, we show that maximizing $\mathbb{E}_{p_d(\mathbf{x})}[\mathcal{L}_{\text{HVAE}}(\mathbf{x})]$ with respect to the prior corresponds to learning the prior by minimizing $\mathbb{E}_{q(\mathbf{z}_{<k})}[\text{KL}(q(\mathbf{z}_k|\mathbf{z}_{<k})||p(\mathbf{z}_k|\mathbf{z}_{<k}))]$ for each conditional:

$$
\begin{aligned}
\arg\max_{p(\mathbf{z}_k|\mathbf{z}_{<k})} \mathbb{E}_{p_d(\mathbf{x})}[\mathcal{L}_{\text{HVAE}}(\mathbf{x})] &= \arg\min_{p(\mathbf{z}_k|\mathbf{z}_{<k})} \mathbb{E}_{p_d(\mathbf{x})}\left[\mathbb{E}_{q(\mathbf{z}_{<k}|\mathbf{x})}[\text{KL}(q(\mathbf{z}_k|\mathbf{z}_{<k},\mathbf{x})||p(\mathbf{z}_k|\mathbf{z}_{<k}))]\right] \\
&= \arg\min_{p(\mathbf{z}_k|\mathbf{z}_{<k})} -\mathbb{E}_{p_d(\mathbf{x})q(\mathbf{z}_{<k}|\mathbf{x})q(\mathbf{z}_k|\mathbf{z}_{<k},\mathbf{x})}[\log p(\mathbf{z}_k|\mathbf{z}_{<k})] \\
&= \arg\min_{p(\mathbf{z}_k|\mathbf{z}_{<k})} -\mathbb{E}_{q(\mathbf{z}_k,\mathbf{z}_{<k})}[\log p(\mathbf{z}_k|\mathbf{z}_{<k})] \\
&= \arg\min_{p(\mathbf{z}_k|\mathbf{z}_{<k})} -\mathbb{E}_{q(\mathbf{z}_{<k})}\left[\mathbb{E}_{q(\mathbf{z}_k|\mathbf{z}_{<k})}[\log p(\mathbf{z}_k|\mathbf{z}_{<k})]\right] \\
&= \arg\min_{p(\mathbf{z}_k|\mathbf{z}_{<k})} \mathbb{E}_{q(\mathbf{z}_{<k})}\left[-H(q(\mathbf{z}_k|\mathbf{z}_{<k})) - \mathbb{E}_{q(\mathbf{z}_k|\mathbf{z}_{<k})}[\log p(\mathbf{z}_k|\mathbf{z}_{<k})]\right] \\
&= \arg\min_{p(\mathbf{z}_k|\mathbf{z}_{<k})} \mathbb{E}_{q(\mathbf{z}_{<k})}[\text{KL}(q(\mathbf{z}_k|\mathbf{z}_{<k})||p(\mathbf{z}_k|\mathbf{z}_{<k}))] .
\end{aligned}
\tag{7}
$$

## C   Conditional NCE for Hierarchical VAEs

In this section, we describe how we derive the NCE training objective for hierarchical VAEs given in Eq. (4). Our goal is to learn the likelihood ratio between the aggregate conditional $q(\mathbf{z}_k|\mathbf{z}_{<k})$ and the prior $p(\mathbf{z}_k|\mathbf{z}_{<k})$. We can define the NCE objective to train the discriminator $D_k(\mathbf{z}_k, \mathbf{z}_{<k})$ that classifies $\mathbf{z}_k$ given samples from the previous groups $\mathbf{z}_{<k}$ using:

$$
\min_{D_k} -\mathbb{E}_{q(\mathbf{z}_k|\mathbf{z}_{<k})}[\log D_k(\mathbf{z}_k, \mathbf{z}_{<k})] - \mathbb{E}_{p(\mathbf{z}_k|\mathbf{z}_{<k})}[\log(1 - D_k(\mathbf{z}_k, \mathbf{z}_{<k}))] \quad \forall \mathbf{z}_{<k}. \tag{8}
$$

Since $\mathbf{z}_{<k}$ is in a high dimensional space, we cannot apply the minimization $\forall \mathbf{z}_{<k}$. Instead, we sample from $\mathbf{z}_{<k}$ using the aggregate approximate posterior $q(\mathbf{z}_{<k})$ as done for the KL in a hierarchical model (Eq. (7)):

$$
\min_{D_k} \mathbb{E}_{q(\mathbf{z}_{<k})}\left[-\mathbb{E}_{q(\mathbf{z}_k|\mathbf{z}_{<k})}[\log D_k(\mathbf{z}_k, \mathbf{z}_{<k})] - \mathbb{E}_{p(\mathbf{z}_k|\mathbf{z}_{<k})}[\log(1 - D_k(\mathbf{z}_k, \mathbf{z}_{<k}))]\right]. \tag{9}
$$

Since $q(\mathbf{z}_{<k})q(\mathbf{z}_k|\mathbf{z}_{<k}) = q(\mathbf{z}_k, \mathbf{z}_{<k}) = \mathbb{E}_{p_d(\mathbf{x})}[q(\mathbf{z}_{<k}|\mathbf{x})q(\mathbf{z}_k|\mathbf{z}_{<k},\mathbf{x})]$, we have:

$$
\min_{D_k} \mathbb{E}_{p_d(\mathbf{x})q(\mathbf{z}_{<k}|\mathbf{x})}\left[-\mathbb{E}_{q(\mathbf{z}_k|\mathbf{z}_{<k},\mathbf{x})}[\log D_k(\mathbf{z}_k, \mathbf{z}_{<k})] - \mathbb{E}_{p(\mathbf{z}_k|\mathbf{z}_{<k})}[\log(1 - D_k(\mathbf{z}_k, \mathbf{z}_{<k}))]\right]. \tag{10}
$$

Finally, instead of passing all the samples from the previous latent variables groups to $D$, we can pass the context feature $c(\mathbf{z}_{<k})$ that extracts a representation from all the previous groups:

$$
\min_{D_k} \mathbb{E}_{p_d(\mathbf{x})q(\mathbf{z}_{<k}|\mathbf{x})}\left[-\mathbb{E}_{q(\mathbf{z}_k|\mathbf{z}_{<k},\mathbf{x})}[\log D_k(\mathbf{z}_k, c(\mathbf{z}_{<k}))] - \mathbb{E}_{p(\mathbf{z}_k|\mathbf{z}_{<k})}[\log(1 - D_k(\mathbf{z}_k, c(\mathbf{z}_{<k})))]\right].
\tag{11}
$$

## D   NVAE Based Model and Context Feature

**Context Feature:** The base model NVAE [74] is hierarchical. To encode the information from the lower levels of the hierarchy to the higher levels, during training of the binary classifiers, we concatenate the context feature $c(\mathbf{z}_{<k})$ to the samples from both $p(\mathbf{z})$ and $q(\mathbf{z})$. The context feature for each group is the output of the residual cell of the top-down model and encodes a representation from $\mathbf{z}_{<k}$.

**Image Decoder $p(\mathbf{x}|\mathbf{z})$:** The base NVAE [74] uses a mixture of discretized logistic distributions for all the datasets but MNIST, for which it uses a Bernoulli distribution. In our model, we observe that replacing this with a Normal distribution for the RGB image datasets leads to significant improvements in the base model performance. This is also reflected in the gains of our approach.

## E   Implementation Details

The binary classifier is composed of two types of residual blocks as in Fig. 6. The residual blocks use batch-normalization [32], the Swish activation function [61], and the Squeeze-and-Excitation (SE)

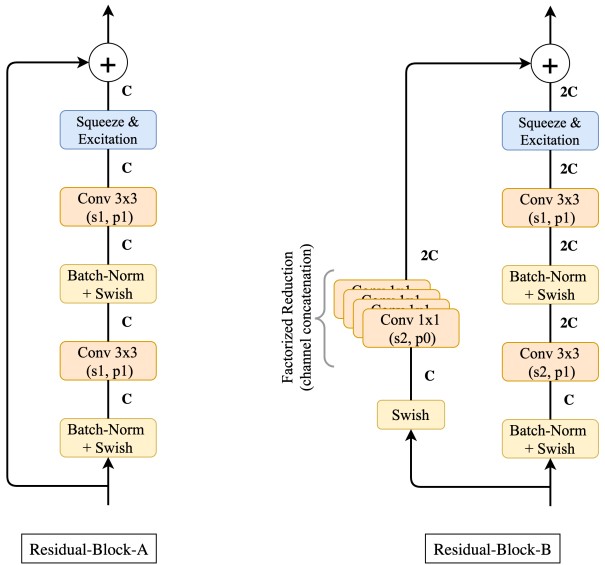

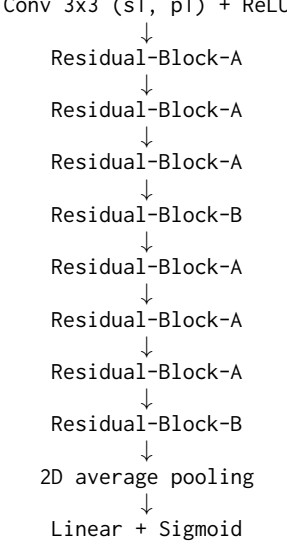

**Figure 6:** Residual blocks used in the binary classifier. We use *s*, *p* and *C* to refer to the stride parameter, the padding parameter and the number of channels in the feature map, respectively.

block [31]. SE performs a *squeeze* operation (*e.g.*, mean) to obtain a single value for each channel. An *excitation* operation (non-linear transformation) is applied to these values to get per-channel weights. The Residual-Block-B differs from Residual-Block-A in that it doubles the number of channels ($C \rightarrow 2C$), while down-sampling the other spatial dimensions. It therefore also includes a factorized reduction with $1 \times 1$ convolutions along the skip-connection. The complete architecture of the classifier is:

```
Conv 3x3 (s1, p1) + ReLU
           ↓
    Residual-Block-A
           ↓
    Residual-Block-A
           ↓
    Residual-Block-A
           ↓
    Residual-Block-B
           ↓
    Residual-Block-A
           ↓
    Residual-Block-A
           ↓
    Residual-Block-A
           ↓
    Residual-Block-B
           ↓
    2D average pooling
           ↓
    Linear + Sigmoid
```

| Optimizer | Adam [37] |
|---|---|
| Learning Rate | Initialize at $1e$-3, CosineAnnealing [49] to $1e$-7 |
| Batch size | 512 (MNIST, CIFAR-10), 256 (CelebA-64), 128 (CelebA HQ 256 ) |

**Table 10:** Hyper-parameters for training the binary classifiers.

# F   Additional Examples - Nearest Neighbors from the Training Dataset

Query Image                                     Nearest neighbors from the training dataset                                     .

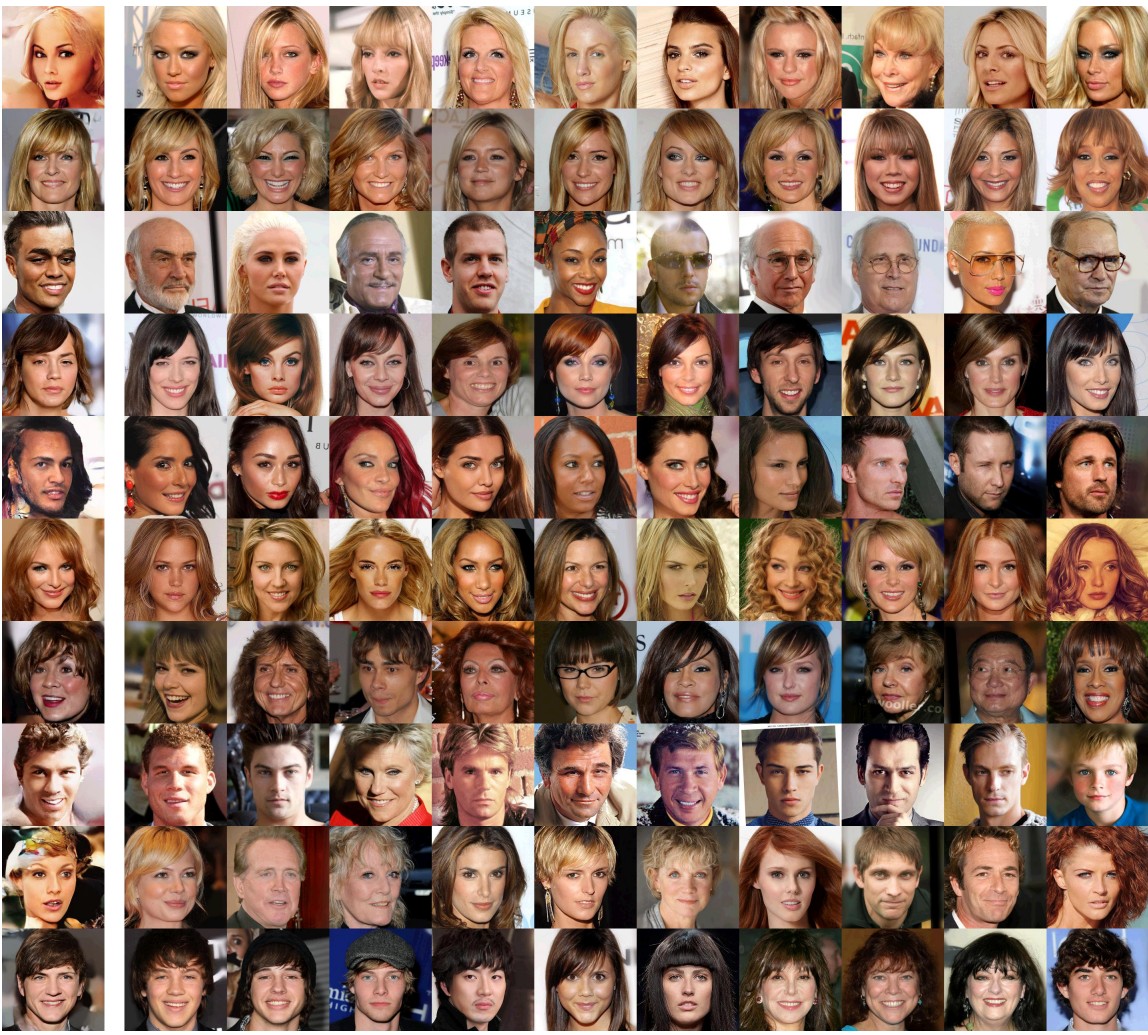

**Figure 7:** Query images (left) and their nearest neighbors from the CelebA-HQ-256 training dataset.

# G  Additional Qualitative Examples

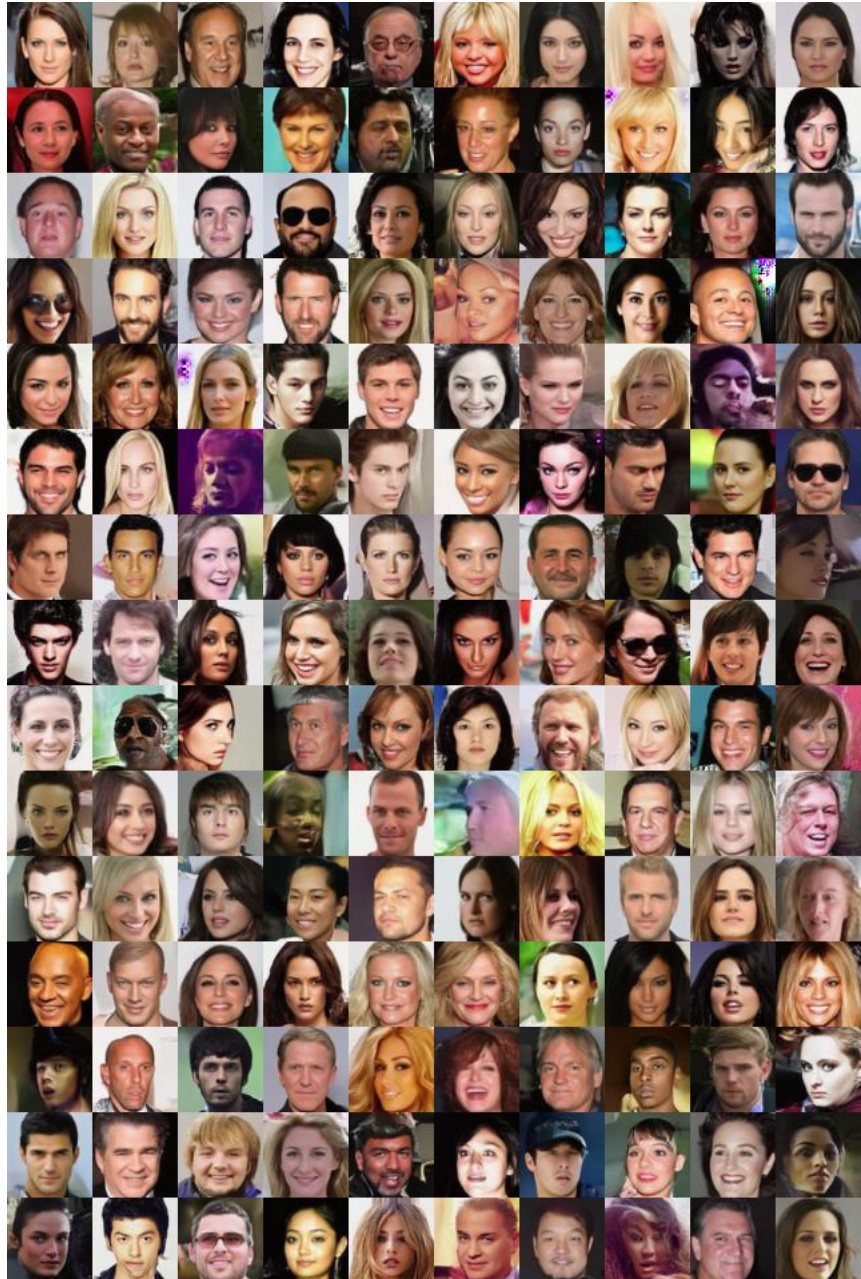

**Figure 8:** Additional samples from CelebA-64 at $t = 0.7$.

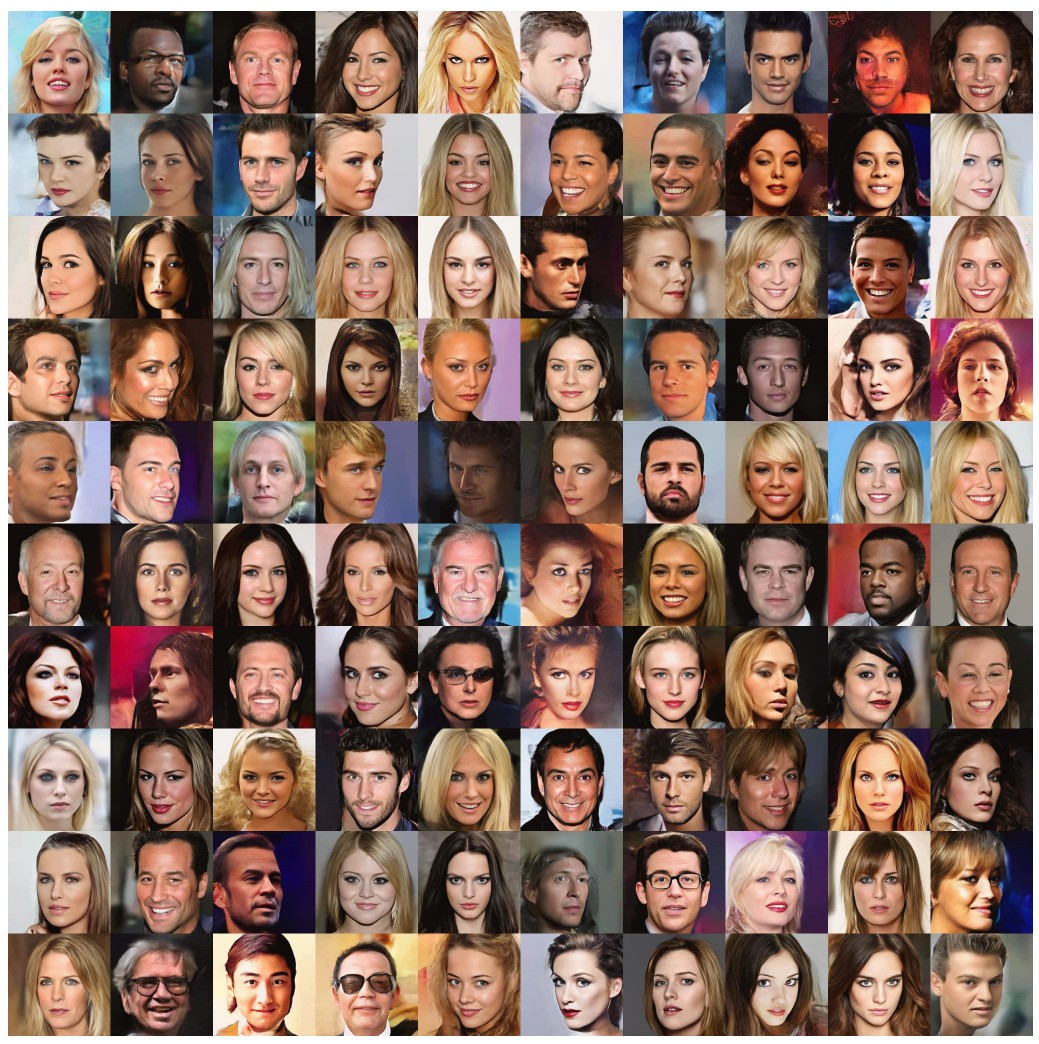

**Figure 9:** Additional samples from CelebA-HQ-256 at $t = 0.7$.

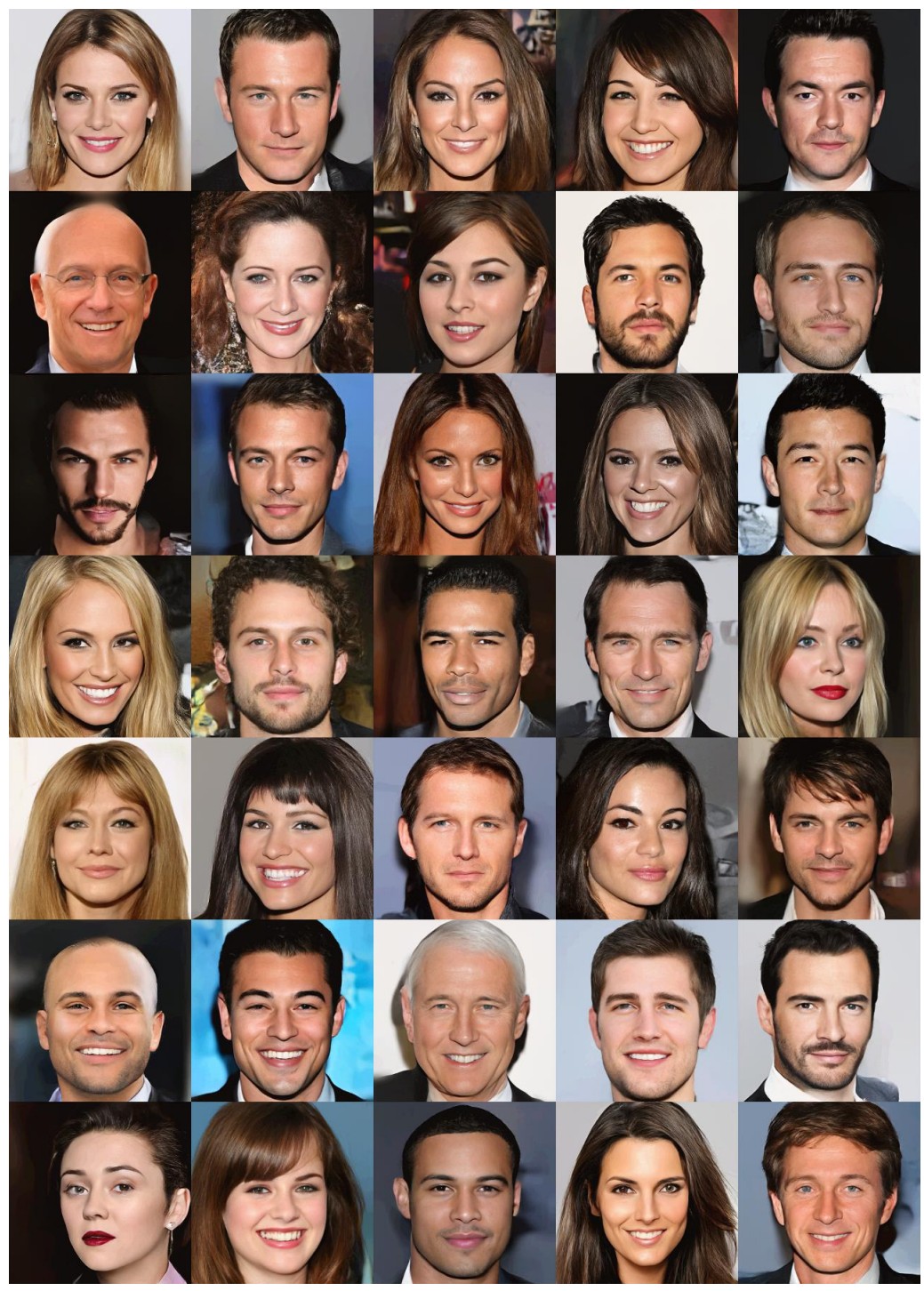

**Figure 10:** Selected good quality samples from CelebA-HQ-256.

# H   Additional Qualitative Examples

In Fig. 11, we show additional examples of images generated by NVAE [74] and our NCP-VAE. We use temperature ($t = 0.7$) for both. Visually corrupt images are highlighted with a red square.

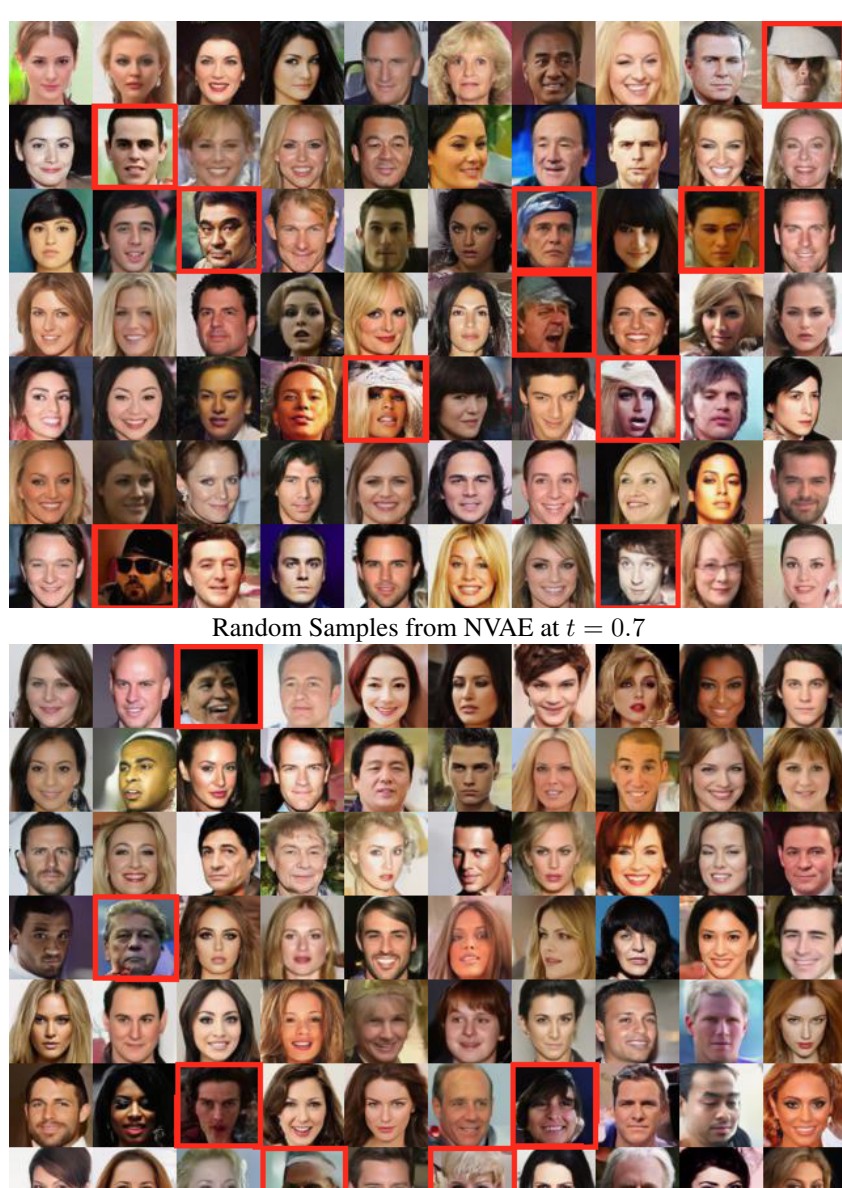

Random Samples from NVAE at $t = 0.7$

Random Samples from NCP-VAE at $t = 0.7$

**Figure 11:** Additional samples from CelebA-64 at $t = 0.7$.

# I  Additional Qualitative Examples

In Fig. 12, we show additional examples of images generated by our NCP-VAE at $t = 1.0$.

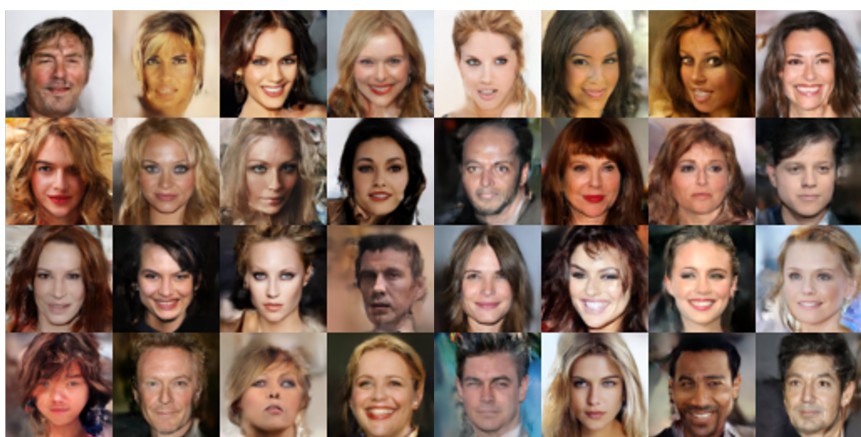

**Figure 12:** Additional samples from CelebA-64 at $t = 1.0$.

# J  Experiment on Synthetic Data

In Fig. 13 we demonstrate the efficacy of our approach on the 25-Gaussians dataset, that is generated by a mixture of 25 two-dimensional Gaussian distributions that are arranged on a grid. The encoder and decoder of the VAE have 4 fully connected layers with 256 hidden units, with 20 dimensional latent variables. The discriminator has 4 fully connected layers with 256 hidden units. Note that the samples decoded from prior p(z) Fig. 13(b)) without the NCP approach generates many points from the the low density regions in the data distribution. These are removed by using our NCP approach (Fig. 13(c)).

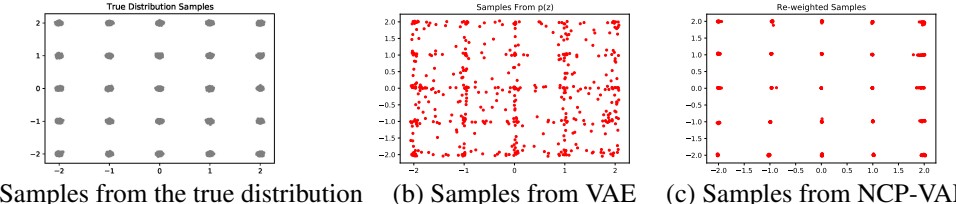

(a) Samples from the true distribution    (b) Samples from VAE    (c) Samples from NCP-VAE

**Figure 13:** Qualitative results on mixture of 25-Gaussians.

We use 50k samples from the true distribution to estimate the log-likelihood. Our NCP-VAE obtains an average log-likelihood of $-0.954$ nats compared to the log-likelihood obtained by vanilla VAE, $-2.753$ nats. We use 20k Monte Carlo samples to estimate the log partition function for the calculation of log-likelihood.