# OpenReview forum: "A Contrastive Learning Approach for Training Variational Autoencoder Priors"
_NeurIPS.cc/2021/Conference — NeurIPS 2021 Poster_

### Official Review · Reviewer_wa7R · 2021-06-30

**Rating:** 7
**Confidence:** 4

**Summary:**

Update after Response:

The response of the authors resolved some of my concerns and I am therefore improving my score by 1 point, provided that the authors add the additional information mentioned in their reply. I think the work is relevant and presents a potential solution to solve some of the problems that plague autoencoder sample generation.

Old:
The authors propose a novel method of adapting the prior p(z) of VAE to the experimentally observed approximate posterior q_{\theta}(z) over the training samples. The modification of the prior is performed as a separate step, after model training. This improves the visual quality of generated samples drawn from the modified prior. The article includes an extensive evaluation and the appendix includes further results with important studies.

**Limitations And Societal Impact:**

The authors do not discuss limitations of their work. An important limitation of their work is that their proposed SIR sampling can fail. By proposing samples using the prior it can be difficult to sample regions which have high probability under the marginal posterior but low probability under the prior. While the authors explore Langevin sampling as alternative, the study does not compare how well the samples drawn by the samples reflect the marginal posterior.

While the impact contains an impact statement, it is only of limited value since it is mostly used to praise the work presented. The societal impact of generating images of high fidelity could include for example the dangers of creating deep fakes. I do appreciate however the reference to bias in image generation.

**Main Review:**

Strengths:
- very clearly written method
- simple, straightforward method
- detailed evaluation on multiple datasets and with multiple baselines
- High visual quality of qualitative samples shown.
- topic highly relevant to the neurips community

Weaknesses:
- implementation not accessible
- On initial reading, I feared that by adapting the prior to the aggregate approximate posterior of the training dataset, we are teaching the model memorization of the training set. The nearest-neighbor evaluation given in Appendix F has convinced me of the opposite. Sad, that I had to reach to the appendix for this very relevant experimental result.
- evaluation by FID: What has been used as the comparison data distribution? If the training set is used, I'm not surprised that this method achieves high FID scores, as the NCE training step is focused on aligning the prior to the approximate posterior of the training dataset. How do the FID metric on the validation/test set compare?
- In the experiments, it is unclear how the log partition function is computed. It is well known that for many estimators, a small standard deviation is not a reliable estimate for correctness of the mean since the distribution of estimates can be highly skewed.
- I feel the use of the t-value during image generation, e.g. Figure 3, undermines the message of the paper. If it is beneficial to accurately represent the marginal posterior, why would we then sample from a different distribution than the marginal posterior?  Also, it is unclear how the temperature was selected.

Correctness:
The experimental evaluation is mostly appropriate to determine the correctness of the work. However, details are unclear.


Clarity: Is the paper well written?
- The introduction and method sections are very clearly written
- The Experimental section could explain the total model size, used compute, and evaluation metrics used in more detail/higher clarity.
- Consider moving required background knowledge for the method from the related work section (after the method) to the background section (before the method).


Additional feedback:

The authors phrase the problem as the inability of the prior to model the posterior correctly. But wouldn't it rather be the inability of the posterior to correctly model the prior or our inability to train a posterior that accurately models the prior? It is well known that GANs provide very good results when sampling from a standard normal prior, thus it should be possible to train a VAE that uses the same prior. I think the authors should provide more evidence to the claim that the prior is at fault here and not the posterior/ELBO or rephrase the relevant parts of the paper.

The checklist includes several answers that are debatable
- 1b) I do not see section 4 discussing limitations of the presented work but only limitations of the presented work. I would expect those to be discussed in the discussion/conclusion of experiments and results.
- 3a) asks for code to which the authors reply yes, without making the code accessible. A description of the method is not equivalent to making the code public, e.g. to check for correctness or reproduction of the experiments.
- 3b) I think number of training iterations/stopping criteria are missing.
- 3d) asks for total estimated compute time used for the experiments. The paper only includes partial measurements and it is unclear how costly a full reproduction of the results would be.


Typos:
line 196, prior art -> prior work
line 199, In most our experiments -> In most of our experiments



**Time Spent Reviewing:**

5h

---

> ### Author Response · Authors · 2021-08-10
> **Response to Reviewer wa7R**
>
> We thank the reviewer for providing valuable and constructive feedback on our manuscript. We address the concerns below.
>
> * **Implementation not accessible**
>
> We have provided the implementation of the code at [Code_6844](https://drive.google.com/drive/folders/15tCGruQcSdm2G4yLkUpKvGASluSZPIBD?usp=sharing). We will release the code and all the trained checkpoints on acceptance.
>
> * **Very relevant nearest neighbor evaluation in supplementary material only**
>
> Thank you for pointing this out. We’ll move the nearest neighbor examples from appendices to the main paper.
>
> * **Validation/test set FID**
>
> Following the standard VAE evaluation, we use 50K reference images for calculating the FID score. Since the size of the val/test set is much smaller than 50K, we, like other VAE papers too, use part of the training set for calculation of the FID. To highlight that our approach generates unseen samples at test time rather than memorizing the training dataset, we visualize samples from the model along with a few training images that are most similar to them (nearest neighbors) in Appendix F. Note that the generated samples are quite distinct from the training images.
>
> * **How is the log-partition-function/NLL computed**
>
> For NCP-VAEs we compute the test log-likehood using Burda et al. [a]’s importance weighted estimation of the test data log-likelihood. However, for this, we also require estimating the likelihood of latent variables under the prior which is intractable. We examine two log-likelihood estimators for the prior: 1) the importance weighted sampling that provides a stochastic upper bound on the log-likelihood; and 2) the bound introduced by Lawson et al. [45] (Eq. 3) that provides a lower bound on the log-likelihood (note that these two bounds are very similar to each other and they differ only by one term). In our experiments on small latent variable spaces on MNIST in Table 3, we observe that these two bounds are sandwiching the true log-likelihood value very tightly. We will provide these additional details on likelihood estimation.
>
> [a] Burda et al., Importance Weighted Autoencoders.
>
> * **How was the temperature selected**
>
> The reported FID scores are calculated without tempering, i.e., we used the temperature $T=1.0$ when computing FID. When visualizing samples, it is a common practice in the VAE community to reduce the temperature of the base prior by scaling down the std-deviation of the normal distribution. The temperature for the illustrated images as mentioned in the corresponding captions,  was chosen empirically by visually inspecting  the quality and smoothness of the images. We’ll add qualitative examples for T=1.0 in the final camera-ready version.
>
>
> * **Experimental details**
>
> The discriminators are trained using the binary cross entropy loss. The final model is chosen after the classifier loss is minimised and the difference between the loss value of subsequent epochs is less than 0.01, which is observed to happen within 100 epochs.
>
> We use the following number of latent variable groups (and number of groups per scale) in the base NVAE for CIFAR-10, CelebA 64, CelebA HQ 256 are 30 (30), 15 (5, 5, 5) and 20 (4, 4, 4, 4, 4), respectively. Correspondingly, the number of  parameters for discriminators for CelebA 64 per scale are 2.3M, 2.4M, and  2.5M. For celebA 256, the number of parameters for discrimnators at each scale are 4.3M, 3.9M, 3.7M, 3.6M, 3.5M and For CIFAR10, Each discriminator has 2.4 M parameters. We use FID [25] as an evaluation metric.
>
> * **Paper structure suggestions**
>
> Thanks for the suggestion, we will move the required background knowledge for the method from the related work section (after the method) to the background section (before the method).
>
> * **Is the prior distribution at fault?**
>
> Let’s consider a scenario in which we could perform perfect sampling from the true posterior hypothetically (i.e., the expressivity of the approximate posterior is no longer an issue). In this case, we will obtain high quality sampling from our VAE only if the decoder can also successfully map samples from the base Normal prior to the complex multi-modal data space. Unfortunately, training such a decoder is very challenging given the small sample size, imperfect optimization, and small networks. That’s why the decoder itself may introduce prior holes as it cannot perfectly map the simple latent space to a complex multi-modal data space.
>
> As you pointed out correctly, GANs are the best known models that perform the mapping from a simple Normal distribution to the data space successfully. However, GANs drop modes in this process and perhaps that’s why they don’t introduce prior holes. Whereas in VAEs, the ELBO objective does not allow the VAE generator to drop any modes.
>
> The NVAE base model used in our work comes with an expressive approximate posterior in the form of hierarchical models and normalizing flows. Yet, we observe a mismatch between the aggregate posterior and the prior (line 286). We believe that even in the case of improved approximate posteriors, due to the capacity and expressivity of the decoder, we still face the prior hole problem.
>
> * **Checklist limitation discussion**
>
> We thank the reviewer for pointing this out and also suggesting some of the limitations. We agree that the perils of creating deep fakes is a great societal challenge. In this work, we explored both SIR and Langevin Dynamics and observed that both work well (see Table 8). We would like to point out that a crucial component for SIR to work successfully is the closeness of the proposal and the target distributions. The base model NVAE has a trainable prior, and for most groups of the hierarchical latent space, the prior distribution is close to the aggregate posterior distribution as reflected in Fig. 4(a). However, if the prior is too different from the aggregate posterior, the SIR sampling may fail. In that case, we can explore more advanced sampling techniques such as Hamiltonian Monte Carlo sampling. We can also define the NCP prior in smaller subsets of the latent space in an autoregressive structure.
>
> * **Checklist compute time**
>
> The discriminator model is trained for 100 epochs per group. It takes 13 hours on a single NVIDIA Tesla V100, 32GB GPU. We use 15 GPU’s to train all the discriminators in parallel. For Celeb 64, it takes 8.124 GPU days, for CIFAR 10 it takes 16.25 GPU days and CelebA 265 it takes 11 GPU days, to train all the discriminators.
>
> * **Typos**
>
> Thanks for pointing out, we will correct the typos in the final version.
>
> **Final remark**: We hope that our response could answer your questions and concerns. If you have additional questions/comments/concerns please feel free to comment here during the discussion period. Otherwise, we would appreciate it if you consider raising your score.

---

### Official Review · Reviewer_C3kP · 2021-07-16

**Rating:** 6
**Confidence:** 3

**Summary:**

The paper proposes to learn an energy based model using noise contrastive estimation on the latent of a VAE to mitigate the prior hole problem. To do so, the paper proposes a very simple method that consists in first training a standard VAE, then, given the frozen VAE, training a discriminator in latent space to discriminate between samples from the prior, and samples from encodings of real images. The discriminator can then be used to reweigh the prior, and obtain a better, but non algebraically computable, prior distribution. The implicit distribution can still be sampled using importance resampling or langevin dynamics. The paper proceeds to show that this technique improves the performance of the model in term of FID on various datasets.

**Limitations And Societal Impact:**

The authors provide a statement assessing the potential societal impact of their work.

**Main Review:**

Pros:
- The idea is very simple and easy to implement.
- The method can be plugged with many different VAE variants.
- The method seems to improve sample quality as measured by FID across the board.
- The paper provides quite thorough empirical validations, on many different datasets, with many different architectures.
- The ablations, while mainly confirming intuitions, are informative.
- The paper is well written, and most claims are well substantiated.

Cons:
- The method cannot reliably be used to evaluate a bound on the log-likelihood.
- The only performance metric used is the FID, which arguably favors quality of samples over generalization (as mentionned in (Razavi et al. 2019). As a consequence, I am not sure that the claim that 'diversity' of samples is improved (l.319) is well substantiated. Evaluating performance using a more diverse set of metrics, and notably the metrics used in (Razavi et al. 2019) could help in substantiating this claim.

Other comments:
- I don't understand the statement l.240. Fig 1. from Kingma et al. 2016 seems to indicate that IAF exactly attend to the prior holes problem.
- l294. Could you also give the number of proposal samples, so that we can compare it to the effective sample size?

Overall, I think the pros outweigh the cons. This is a simple and interesting contribution. I recommend acceptance.

References:
_Generating Diverse High-Fidelity Images with VQ-VAE-2_ Ali Razavi, Aäron van den Oord, Oriol Vinyals

**Time Spent Reviewing:**

3

---

> ### Author Response · Authors · 2021-08-10
> **Response to Reviewer C3kP**
>
> We thank the reviewer for providing valuable and constructive feedback on our manuscript. We address the concerns below.
>
> * **Reliable estimation of log-likelihoods**
>
> For NCP-VAEs we compute the test log-likehood using Burda et al. [a]’s importance weighted estimation of the test data log-likelihood. However, for this, we also require estimating the likelihood of latent variables under the prior, which is intractable. We examine two log-likelihood estimators for the prior: 1) the importance weighted sampling that provides a stochastic upper bound on log-likelihood; and 2) the bound introduced by Lawson et al. [45] (Eq. 3) that provides a lower bound on the log-likelihood (note that these two bounds are very similar to each other and they differ only by one term). In our experiments on small latent variable spaces on MNIST in Table 3, we observe that these two bounds are sandwiching the true log-likelihood value very tightly. We will provide these additional details on likelihood estimation.
>
> * **Additional metrics by Razavi et al. (2019)**
>
> We report FID for all the datasets and NLL for MNIST. The additional metrics reported by Razavi et al. (2019), i.e., “Precision-Recall Metric” and “Classification Accuracy Score” require labels. Note that we are operating in an unconditional setting, i.e., no image-labels are available. Hence we cannot trivially compute the metrics suggested by Razavi et al. (2019).
>
> * **Justify that diversity is improved**
>
> NLL on the test dataset is sensitive to mode dropping. The reported NLL values for MNIST on two architectures (table 2 and 6) and for the toy dataset (Appendix H) do not show any sign of mode dropping. Additionally, to highlight that our approach generates unseen samples at test time rather than memorizing the training dataset, in Appendix F, we show samples from the model along with a few training images that are most similar to them (nearest neighbors). Note that the generated samples are quite distinct from the training images.
>
> * **Clarify statement in l.240**
>
> Kingma et al. (2016) show that techniques like hierarchical latent spaces and normalizing flow in the posterior are intuitively expected to reduce the prior hole problem. Nonetheless, for large-scale models we find that the prior hole problem still exists. Specifically, the base model NVAE uses both normalizing flows and hierarchical distributions, yet the prior hole problem remains as is evident from our experiments.
>
> * **Number of proposal samples**
>
> We use 100 proposals for effective sample size calculation.
>
>
> **Final remark**: We hope that our response could answer your questions and concerns. If you have additional questions/comments/concerns please feel free to comment here during the discussion period. Otherwise, we would appreciate it if you consider raising your score.

---

### Official Review · Reviewer_DUyg · 2021-07-26

**Rating:** 6
**Confidence:** 4

**Summary:**

In trained VAEs, mismatch between the prior and the variational posterior can lead to low-quality generated samples.

To address this issue, the authors suggest the following procedure:
1) given a trained VAE, train a classifier to distinguish images generated from the variational posterior vs images generated from the prior;
2) adjust the prior to minimize the accuracy of the trained classifier from step 1. This is done by reweighting the sampling procedure using the predictions from the classifier.

Main contribution:
a practical method to improve the generative quality of several forms of VAE by reweighing the prior to diminish the mismatch from the variational posterior

**Main Review:**

Originality:
Overall, the combination of ideas/methods to improve image generation in VAEs is original.

Quality:
The submission is technically sound.

Clarity:
Generally well written. I have two minor comments:
- I think the manuscript would read better if the authors first discuss the main steps of the procedure (train classifier given a trained VAE, then inference with sampling reweighting) as done in Section 3.2 and then describe the specifics of contrastive estimation in Section 3.1;
- the authors state that prior hole problem is when "the prior distribution fails to match the aggregate approximate posterior". I would say that the contrary happens: after training, the aggregate approximate posterior may not match the prior sufficiently well (this may due to limited expressivity of the posterior, trainability, low sample size, etc);

Significance:
This is my only major concern. While I see the practicality of this procedure, the technical contribution is narrow and I wonder why one would prefer this procedure over applying state-of-the-art VAE methods such as VDVAE and VQVAE. Could the authors provide some guidance on the use cases where their method should be preferred?

After revision:
After reading authors' reply and other reviewers' comments, I decided to change my score to a 6 (+2 difference from previous score). I think my original assessment on significance was incomplete. As Reviewer C3kP notices (i) "the idea is very simple and easy to implement, and "The method can be plugged with many different VAE variants". After more consideration, these pros address my original concern on the significance of this work. The authors may consider making this more explicit in their manuscript, should it be accepted.

**Time Spent Reviewing:**

4

---

> ### Author Response · Authors · 2021-08-10
> **Response to Reviewer DUyg**
>
> We thank the reviewer for their comments and suggestions on our manuscript. Below, we address the concerns in detail.
>
> * **Reordering of the sections**
>
> Thanks for the great suggestion about reordering the method subsections for better presentation. We acknowledge that this would make our work easier to understand. We’re happy to apply this reordering in the final version.
>
> * **Prior not matching aggregate posterior or aggregate posterior not matching prior**
>
> There has been tremendous work in the VAE community on making the approximate posterior more flexible. Our base NVAE is equipped with some of these techniques which includes hierarchical distributions and normalizing flows in the posterior. Yet, we observe that there is still a mismatch between the aggregate posterior and the prior after training (line 286 in the submission). Our goal in this paper is to go beyond the expressivity of approximate posteriors and make the prior more expressive by introducing noise-contrastive priors. Our results indicate that the proposed prior does indeed reduce this mismatch, as we observe improved sample quality.
>
> You are raising an interesting point on whether we should only focus on the expressivity and training of the approximate posterior, instead of the prior. Let’s consider a scenario in which we could perform perfect sampling from the true posterior hypothetically (i.e., the expressivity of the approximate posterior is no longer an issue). In this case, we will obtain high quality sampling from our VAE only if the decoder can also successfully map samples from the base Normal prior to the complex multi-modal data space. Unfortunately, training such a decoder itself is very challenging given the small sample size, imperfect optimization, and small networks. That’s why the decoder itself may introduce prior holes as it cannot perfectly map the simple latent space to the complex data space. Hence, we believe that even in the case of improved approximate posteriors, due to the capacity and expressivity of the decoder, we may still face the prior hole problem.
>
>
> * **Guidance on whether to use VD-VAE and VQ-VAE**
>
> Our primary goal in this submission is to increase the expressivity of the prior in VAEs. In doing so, we are aiming for an approach that is scalable and can be easily applied to large VAEs. The approaches proposed in VD-VAE and VQ-VAE are orthogonal to our approach in this paper.
>
> Specifically, VD-VAE trains an extremely deep VAE. That’s why it requires 560 GPU days (2.5 weeks on 32 V100 GPUs) for training on 256x256 images. In contrast, NVAE requires 94 GPU days and our introduced NPC-VAE requires only an additional 11 GPU days (i.e., 0.54 day training for each latent variable group, done in parallel for 20 groups). The overall training cost of the NVAE+NCP prior is much lower than the VD-VAE model. In addition, we believe that our approach, with a modest additional computation, can also improve the generative quality of VD-VAE by closing the gap between the prior and the posterior distributions. This is because VD-VAE’s hierarchical structure is similar to NVAE's structure.
>
> VQ-VAE learns discrete latent representations. Our approach can also be applied to VQ-VAE to improve the expressivity of the prior in this model (Importance sampling can be applied to discrete variables too). However, the main practical difficulty of using VQ-VAE for generation is that this model defines an autoregressive distribution over a latent variable of the size 128x128 (=16K variables). Sampling from such a large number of variables sequentially can be very slow in practice. The goal of our submission is to explore alternative expressive priors beyond autoregressive models.
>
>
> **Final remark**: We hope that our response could answer your questions and concerns. If you have additional questions/comments/concerns please feel free to comment here during the discussion period. Otherwise, we would appreciate it if you consider raising your score.

---

### Official Review · Reviewer_a91M · 2021-07-26

**Rating:** 7
**Confidence:** 4

**Summary:**

The authors propose a way to improve the performance of a trained VAE by modifying the learned prior with a reweighting factor. The reweighting factor is computed using a network trained to discriminate between samples from the learned prior and samples from the encoder given real data. To sample from the VAE, the latent variables are sampled from the reweighted prior using MC inference (e.g. sampling-importance-resampling) and fed through the decoder. This leads to better samples (versus not using reweighting) but also greater computational cost due to the MC inference.

**Limitations And Societal Impact:**

Yes.

**Main Review:**

# Novelty
The authors situate their work well in relation to prior work. Other (cited) papers use reweighted priors, so the novelty comes mainly from using noise contrastive estimation to train the reweighting factor (via learning to discriminate between samples from the prior and from the encoder). This is a significant improvement as it makes training simpler and scalable to larger VAE architectures.

# Strengths
- Clearly written and a pleasure to read.
- Thorough experimental comparisons and ablation studies.
- Applicable to modern hierarchical VAE architectures, unlike much prior work.

# Concerns
- Sampling from the VAE with the suggested parameters (e.g. using 5000 proposal samples) is very slow, taking 10s to create a single output (based on Table 8). This might limit the practical applicability of this approach. It would also be interesting to see these times compared to those for the VAEBM baseline, which also involves MC inference.
- The experimental results are not overly convincing, with the reported FIDs/NLLs often being similar to, or worse than, the baselines. However, given that the authors address this (e.g. pointing out that VAEBM baseline is more complex and not applicable to discrete data), I think the results are good enough for acceptance.

**Time Spent Reviewing:**

4

---

> ### Author Response · Authors · 2021-08-10
> **Response to Reviewer a91M**
>
> We thank the reviewer for providing positive feedback on our manuscript.
>
> * **Time comparison to VAEBM**
>
> Since the sampling involves sequentially drawing M proposals from each level of the
> hierarchy, our approach requires 1.25 seconds to generate an image on CelebA 64 when M = 500. For the same dataset and the same base NVAE model, VAEBM with 20 steps for MCMC requires 1.09 seconds to generate an image which is comparable to our NCP-VAE. Note that using SIR of 5000 proposals, our method generates an image in 10.11 seconds. We observe that there is only a small improvement in FID from 6.76 to 5.25 when increasing the number of proposals by a factor of 10.

---

### Decision · Program_Chairs · 2021-09-27

**Decision:**

Accept (Poster)

**Comment:**

The paper proposes a new class of priors for Variational Auto-Encoders (VAEs). The main idea is to use Noise Contrastive Estimation and a two-stage training method. The reviewers highlighted that:
- The paper is clearly written.
- The idea is simple (in the positive sense!) and easy to implement.
- The proposed prior could be plugged in any VAE.

However, the reviewers raised some concerns:
- Sampling from the VAE with the suggested parameters is very slow.
- The experimental results are not overly convincing.
- The organization of the paper could be improved.
- The method cannot reliably be used to evaluate a bound on the log-likelihood.
- The only performance metric used is the FID, which arguably favors the quality of samples overgeneralization.

Moreover, I find it rather surprising that the authors do not compare their approach to at least one of the following two methods:
- Tomczak, J., & Welling, M. (2018). VAE with a VampPrior. In International Conference on Artificial Intelligence and Statistics (pp. 1214-1223). PMLR.
- Norouzi, S., Fleet, D. J., & Norouzi, M. (2020). Exemplar VAE: Linking Generative Models, Nearest Neighbor Retrieval, and Data Augmentation. arXiv preprint arXiv:2004.04795.

It seems rather natural to focus on a thorough comparison against other priors for VAE. Instead, the authors have a limited comparison in Table 2. Moreover, the authors decided to compare their method to various deep generative models. I totally agree that it is also important, however, without a proper comparison against various priors, it is hard to properly evaluate their idea.

The authors provided a thorough rebuttal and they promised to improve the paper. Overall, the paper is solid and the idea is neat, therefore, I tend to accept the paper.